# 1 A full-coverage satellite-based global atmospheric CO<sub>2</sub> dataset at 0.05°

# 2 resolution from 2015 to 2021 for exploring global carbon dynamics

- 3 Zhige Wang<sup>1,2,3</sup>, Ce Zhang<sup>4,5</sup>, Kejian Shi<sup>6</sup>, Yulin Shangguan<sup>1,2</sup>, Bifeng Hu<sup>7,8</sup>, Xueyao Chen<sup>1,2,9</sup>, Danqing
- Wei<sup>1,2,10</sup>, Songchao Chen<sup>1,2,11\*</sup>, Peter M. Atkinson<sup>12,13</sup>, Qiang Zhang<sup>3</sup>
- <sup>1</sup> State Key Laboratory of Soil Pollution Control and Safety, Zhejiang University, Hangzhou, 310058,
- 6 China
- <sup>7</sup> College of Environmental and Resource Sciences, Zhejiang University, Hangzhou 310058, China
- <sup>3</sup> Ministry of Education Key Laboratory for Earth System Modeling, Department of Earth System Science,
- Tsinghua University, Beijing 100084, China
- <sup>4</sup> School of Geographical Sciences, University of Bristol, Bristol BS8 1SS, UK
- 5 UK Centre for Ecology & Hydrology, Library Avenue, Bailrigg, Lancaster LA1 4AP, UK
- <sup>6</sup> School of Geographical Sciences, Faculty of Science and Engineering, University of Nottingham
- Ningbo China, Ningbo 315100, China
- <sup>7</sup> Department of Land Resource Management, School of Public Administration, Jiangxi University of
- Finance and Economics, Nanchang, 330013, China
- <sup>8</sup>Key Laboratory of Data Science in Finance and Economics of Jiangxi Province, Jiangxi University of
- Finance and Economics, Nanchang, 330013, China
- <sup>9</sup>Department of Earth and Environmental Sciences, Faculty of Science, The Chinese University of Hong
- Kong, Sha Tin, Hong Kong, China
- <sup>10</sup> Zhejiang Economic Information Center, Hangzhou, 310006, China
- II ZJU-Hangzhou Global Scientific and Technological Innovation Center, Zhejiang University, Hangzhou
- 311215, China
- Lancaster Environment Centre, Lancaster University, Lancaster LAI 4YO, UK
- <sup>13</sup> Geography and Environment, University of Southampton, Highfield, Southampton SO17 1BJ, UK

25

\* Corresponding author: Songchao Chen (chensongchao@zju.edu.cn)

# 28 Abstract

| The irreversible trend for global warming underscores the necessity for accurate                         |
|----------------------------------------------------------------------------------------------------------|
| monitoring and analysis of atmospheric carbon dynamics on a global scale. Carbon                         |
| satellites hold significant potential for atmospheric CO2 monitoring. However, existing                  |
| studies on global $CO_2$ are constrained by coarse resolution (ranging from $0.25^\circ$ to $2^\circ$ )  |
| and limited spatial coverage. In this study, we developed a new global dataset of                        |
| column-averaged dry-air mole fraction of $CO_2$ (XCO <sub>2</sub> ) at $0.05^\circ$ resolution with full |
| coverage using carbon satellite observations, multi-source satellite products, and an                    |
| improved deep learning model. We then investigated changes in global atmospheric                         |
| $CO_2$ and anomalies from 2015 to 2021. The reconstructed $XCO_2$ products show a better                 |
| agreement with Total Carbon Column Observing Network (TCCON) measurements,                               |
| with R <sup>2</sup> of 0.92 and RMSE of 1.54 ppm. The products also provide more accurate                |
| information on the global and regional spatial patterns of XCO2 compared to origin                       |
| carbon satellite monitoring and previous $\text{XCO}_2$ products. The global pattern of $\text{XCO}_2$   |
| exhibited a distinct increasing trend with a growth rate of 2.32 ppm/year, reaching                      |
| 414.00 ppm in 2021. Globally, XCO <sub>2</sub> showed obvious spatial variability across                 |
| different latitudes and continents. Higher XCO2 concentrations were primarily                            |
| observed in the Northern Hemisphere, particularly in regions with intensive                              |
| anthropogenic activity, such as East Asia and North America. We also validated the                       |
| effectiveness of our XCO2 products in detecting intensive CO2 emission sources. The                      |
| $XCO_2$ dataset is publicly accessible on the Zenodo platform at                                         |
| https://doi.org/10.5281/zenodo.12706142 (Wang et al., 2024a). Our products enable                        |
| enhanced ability in identifying regional- and county-level XCO2 hotpots, carbon                          |
| emissions and fragmented carbon sinks, providing a robust basis for targeted global                      |
| carbon governance policies.                                                                              |

- **Keywords:** Atmospheric carbon dioxide; Satellite carbon monitoring; Deep learning;
- 55 OCO-2/3

# 1. Introduction

Carbon dioxide (CO<sub>2</sub>) is a primary greenhouse gas (GHG). Anthropogenic activities and land use change since the industrial revolution have led to a marked

increase in atmospheric CO<sub>2</sub>, which is widely considered to be a major contributor to climate change, reaching a record-high of 414.71 parts per million (ppm) in 2021 (Friedlingstein et al., 2022). The damaging global climate change caused by atmospheric increases in CO<sub>2</sub> is severe and irreversible (IPCC, 2023; Kemp et al., 2022; Solomon et al., 2009). Consequently, the Paris Agreement announced to hold "the increase in the global average temperature to well below 2°C above pre-industrial levels" and pursue efforts "to limit the temperature increase to 1.5°C above pre-industrial levels." It was also determined that the joined parties should submit their nationally determined contributions (NDCs) to reduce CO<sub>2</sub> emissions.

Accurate monitoring of atmospheric CO<sub>2</sub> concentrations is crucial for measuring global CO<sub>2</sub> emissions mitigation as well as characterizing terrestrial carbon change. Currently, ground-based and airborne platform-based atmospheric CO<sub>2</sub> observation networks, such as the Total Carbon Column Observing Network (TCCON, https://tccondata.org/), are capable of providing CO<sub>2</sub> measurements with high accuracy (Petzold et al., 2016; Wunch et al., 2011, 2010). However, these observation networks are insufficient to fully explore the spatiotemporal patterns of atmospheric CO<sub>2</sub> at a global scale. The launch of a series of carbon observation satellites in recent years has provided favorable opportunities for continuous and large-scale atmospheric CO<sub>2</sub> observation (Buchwitz et al., 2015; Hammerling et al., 2012). The Scanning Imaging Absorption Spectrometer for Atmospheric Chartography (SCIAMACHY) onboard EnviSat was one of the first instruments to monitor the atmospheric column-averaged dry-air mole fraction of CO<sub>2</sub> (XCO<sub>2</sub>) (Bovensmann et al., 1999). The Greenhouse Gases Observing Satellite (GOSAT) launched by Japan utilized the Thermal And Near-Infrared Sensor for carbon Observation (TANSO) instrument to monitor XCO<sub>2</sub> globally, providing products with a spatial resolution of 10 km every three days (Butz et al., 2011). The Orbiting Carbon Observatory-2 (OCO-2) and OCO-3 launched by NASA provide XCO<sub>2</sub> measurements at a finer spatial resolution (Eldering et al., 2017). These sensors are considered among the best for XCO2 observation, featuring larger overlapping swaths that cover areas of ~20×80 km<sup>2</sup> and exhibiting the least retrieval absolute bias, measuring less than 0.4 ppm (Eldering et al., 2019; Taylor et al., 2020). However, the narrow swath of the sensor can only cover limited spatial areas, and caused by the cloud and aerosol contaminations, the data from OCO-2/3 always contain large amount of missing values (Taylor et al., 2016; Crisp et al., 2017). These limitations obstacle the better understanding of the atmosphere-land carbon cycle over large spatial

scale based on satellite observation.

Consequently, several studies have concentrated on generating spatially continuous XCO<sub>2</sub> products based on satellite observations (He et al., 2022; Siabi et al., 2019; Zhang and Liu, 2023). One potential solution is the application of diverse interpolation methods (He et al., 2020; Zeng et al., 2014). However, their results encounter large uncertainty in regions with sparse data coverage, due to the coarse spatial resolution of the original data. In addition, data fusion techniques have gained recognition as an effective method for obtaining full-coverage XCO<sub>2</sub> data (Sheng et al., 2022; He et al., 2022; Siabi et al., 2019; Zhang and Liu, 2023). These techniques can be broadly categorized into two groups. The first category leverages the spatiotemporal correlation inherent in multi-source XCO2 data, fusing them based on this spatiotemporal information (Wang et al., 2023; Sheng et al., 2022). For instance, Wang et al. (2023) introduced a spatiotemporal self-supervised fusion model and generate seamless global XCO<sub>2</sub> data at a spatial resolution of 0.25°. The second category is regression-based methods, which aim to fill the gap by capturing the nonlinear relationship between multi-source XCO<sub>2</sub> measurements and related covariates (He et al., 2022; Siabi et al., 2019; Zhang and Liu, 2023). The specific methodologies include traditional statistical models, geostatistical models and machine learning models. Siabi et al. (2019) employed the Artificial Neural Network (ANN) to establish correlation between XCO<sub>2</sub> and eight environmental variables. Zhang and Liu (2023) utilized the convolution neural networks (CNN) coupled with attention mechanisms to produce full-coverage XCO<sub>2</sub> data across China. Recently, Zhang et al. (2023) developed high spatial resolution global CO<sub>2</sub> concentration data based on deep forest model and multisource satellite products.

Although the development of CO<sub>2</sub> observation satellites and the application of machine learning methods have significantly improved the estimation accuracy of XCO<sub>2</sub>, current studies still face several limitations. Firstly, due to the sparse distribution of satellite XCO<sub>2</sub> data, previous studies always relied on assimilation and reanalysis XCO<sub>2</sub> data, such as CAMS XCO<sub>2</sub> with coarse spatial resolution (0.75°). This reliance often results in final products that closely mirror the assimilation and reanalysis results, leading to an oversmoothed distribution that undermines the high-resolution advantages of satellite data. Furthermore, most current studies estimated the spatial distribution of CO<sub>2</sub> primarily based on vegetation and meteorological information, with limited

consideration of the impact of human activities and emissions, despite these have significant influence on atmospheric CO<sub>2</sub> variability. This limitation also led to estimation results that fail to adequately capture the impact of anthropogenic emissions on atmospheric CO<sub>2</sub>. In addition, most studies that employ regression models to estimate full-coverage XCO<sub>2</sub> are limited to regional or national scales due to the weak transferability of these models. Only a few studies (Zhang et al., 2023) have explored global-scale CO<sub>2</sub> estimation using machine learning approaches, highlighting the need for further research to enhance model generalizability and scalability. Therefore, we intent to develop the global full-coverage XCO<sub>2</sub> products with the capacity to capture both large-scale patterns and fine spatial details. This development leveraged satellite carbon monitoring, multi-source high spatial resolution auxiliary variables and advanced methods that exhibit spatiotemporal transferability to overcome the aforementioned limitations.

In this study, we leveraged time-series OCO-2/3 XCO<sub>2</sub> data and various related environmental variables from multi-source satellites to generate global full-coverage XCO<sub>2</sub> products. The advanced deep learning method was adopted to model time-series XCO<sub>2</sub> and incorporate terrestrial flux, anthropogenic flux and climatic impacts into the parameterization process. These products are designed to meet the following criteria: (1) high validated accuracy to ensure the reliability of the estimates, (2) high spatial resolution capable of capturing fine-scale variations in CO<sub>2</sub> concentrations, and (3) global full-coverage that overcomes missing values in satellite carbon observations. Our XCO<sub>2</sub> products achieved full global coverage with a spatial resolution of 0.05° and a monthly temporal resolution from 2015 to 2021. We also validated our XCO<sub>2</sub> products against in-situ XCO<sub>2</sub> data and other XCO<sub>2</sub> products. Based on our high-resolution products, we explored the spatial and temporal pattern of atmospheric CO<sub>2</sub> globally and identified regions with intense CO<sub>2</sub> emission. Our findings aim to enhance the understanding of carbon dynamics on a global scale through data reconstruction and analysis.

### 2. Materials and methods

In this study, we utilized Google Earth Engine (GEE) to integrate OCO-2/3 XCO<sub>2</sub> data and multiple environmental variables as data inputs. In addition, the attention-based Bidirectional Long Short-Term Memory (At-BiLSTM) model was trained for

building the relationship between OCO-2/3 XCO<sub>2</sub> and the related environmental variables. Then, we reconstructed the global monthly XCO<sub>2</sub> and validated the accuracy of the products against TCCON XCO<sub>2</sub> data and the original OCO-2/3 XCO<sub>2</sub> data. We also analyzed the spatial and temporal variation of XCO<sub>2</sub> over the globe and detect the intense CO<sub>2</sub> emission regions. The methodology framework is shown in Fig.1.

**Figure 1.** The workflow for mapping and exploring global XCO<sub>2</sub> dynamics and drivers.

## 2.1 Datasets

# 2.1.1 OCO XCO<sub>2</sub> data

In this study, we utilized the satellite-based XCO<sub>2</sub> data from OCO-2 and OCO-3, covering the period from December 2014 to December 2021. The OCO-2/3 measure at three near-infrared wavelength bands, that are 0.76  $\mu$ m Oxygen A-band, 1.61  $\mu$ m weak CO<sub>2</sub>, and 2.06  $\mu$ m strong CO<sub>2</sub> bands (Crisp et al., 2004). The full physics retrieval

algorithm was used to retrieve the XCO<sub>2</sub> based on the observation of the two satellites (Crisp et al., 2021). Previous studies (Taylor et al., 2023) suggested that the OCO-2 and OCO-3 XCO<sub>2</sub> measurements are in broad consistency and can therefore be used together in scientific analyses. The OCO-3 Level 2 XCO<sub>2</sub> Lite version 10.4r data (OCO3\_L2\_Lite\_FP V10.4r) from 2020 to 2021 and the OCO-2 Level 2 XCO<sub>2</sub> Lite version 11r (OCO2\_L2\_Lite\_FP V11r) from 2015 to 2019 were downloaded from Goddard Earth Sciences Data and Information Services Center (GES DISC, https://disc.gsfc.nasa.gov/). The products were aggregated as a daily file (Fig. 2) with a spatial resolution of 2.25 km × 1.29 km (O'Dell et al., 2018). The XCO<sub>2</sub> data were quality filtered, and only good-quality data (i.e., xco2\_quality\_flag=0) were considered. To generate the monthly products with a spatial resolution of 0.05° × 0.05°, we converted the daily data to monthly data by averaging the sparse XCO<sub>2</sub> data within a range of 0.05° × 0.05° over one month.

**Figure 2.** Footprints of OCO-2 and OCO-3 XCO<sub>2</sub> data on 20th January 2018 and 4th December 2021 (with quality filtering) as examples.

## 2.1.2 TCCON XCO<sub>2</sub> data

The Total Carbon Column Observing Network (TCCON) is a global network for measuring atmospheric CO<sub>2</sub>, methane (CH<sub>4</sub>), carbon monoxide (CO) and other trace gases in the atmosphere. The XCO<sub>2</sub> data from TCCON were demonstrated to have high accuracy with ~0.2% of XCO<sub>2</sub> (Wunch et al., 2011). Consequently, the data have been used widely for the validation of satellite observations such as OCO-2, OCO-3 and GOSAT (Deng et al., 2016; Wunch et al., 2017). In this research, we used the GGG2014 and GGG2020 datasets from 23 sites (Fig. 3 and Table 1) around the world to validate the reconstructed XCO<sub>2</sub> products.

198199

Figure 3. The locations of the TCCON sites.

200201

**Table 1**. The information on the TCCON *in situ* stations.

| Table 1. The information on the Teeory in still stations. |                      |           |          |            |            |  |  |
|-----------------------------------------------------------|----------------------|-----------|----------|------------|------------|--|--|
| ID                                                        | Site name            | Longitude | Latitude | Start date | End date   |  |  |
| 1                                                         | saga01 (JP)          | 130.29    | 33.24    | 2011-07-28 | 2021-06-30 |  |  |
| 2                                                         | xianghe01 (PRC)      | 116.96    | 39.80    | 2018-06-14 | 2022-04-09 |  |  |
| 3                                                         | burgos01 (PH)        | 120.65    | 18.53    | 2017-03-03 | 2021-08-20 |  |  |
| 4                                                         | harwell01 (UK)       | -1.32     | 51.57    | 2021-05-30 | 2022-05-22 |  |  |
| 5                                                         | bremen01 (DE)        | 8.85      | 53.10    | 2009-01-06 | 2021-06-24 |  |  |
| 6                                                         | tsukuba02 (JP)       | 140.12    | 36.05    | 2014-03-28 | 2021-03-31 |  |  |
| 7                                                         | lauder03 (NZ)        | -97.49    | 36.60    | 2018-10-02 | 2022-11-14 |  |  |
| 8                                                         | edwards01 (US)       | -117.88   | 34.96    | 2013-07-20 | 2022-12-25 |  |  |
| 9                                                         | nicosia01 (CY)       | 33.38     | 35.14    | 2019-09-06 | 2021-06-01 |  |  |
| 10                                                        | izana01 (ES)         | -16.5     | 28.31    | 2014-01-02 | 2022-10-31 |  |  |
| 11                                                        | orleans01 (FR)       | 2.11      | 47.96    | 2009-09-06 | 2022-04-24 |  |  |
| 12                                                        | hefei01 (PRC)        | 119.17    | 31.90    | 2015-11-02 | 2020-12-31 |  |  |
| 13                                                        | easttroutlake01 (CA) | -104.99   | 54.35    | 2016-10-03 | 2022-08-13 |  |  |
| 14                                                        | karlsruhe01 (DE)     | 8.44      | 49.10    | 2014-01-15 | 2023-01-20 |  |  |
| 15                                                        | paris01 (FR)         | 2.36      | 48.85    | 2014-09-23 | 2022-03-28 |  |  |
| 16                                                        | garmisch01 (DE)      | 11.06     | 47.48    | 2007-07-18 | 2021-10-18 |  |  |
| 17                                                        | rikubetsu01 (JP)     | 143.77    | 43.46    | 2014-06-24 | 2021-06-30 |  |  |
| 18                                                        | lamont01 (US)        | 169.68    | -45.04   | 2011-04-16 | 2022-12-19 |  |  |
| 19                                                        | reunion01 (RE)       | 55.48     | -20.90   | 2015-03-01 | 2020-07-18 |  |  |
| 20                                                        | darwin01 (AU)        | 130.93    | -12.46   | 2005-08-28 | 2020-04-30 |  |  |
| 21                                                        | Wollongong (AU)      | 150.88    | -34.41   | 2008-06-26 | 2020-06-30 |  |  |
| 22                                                        | Manaus01(BR)         | -60.60    | -3.21    | 2014-09-30 | 2015-07-27 |  |  |
| 23                                                        | parkfalls01 (US)     | -90.27    | 45.94    | 2004-06-02 | 2020-12-29 |  |  |

JP: Japan, DE: Germany, FI: Finland, FR: French, RE: Réunion Island, AU: Australia, BR: Brazil; US: United States, PRC: People's Republic of China, NO: Norway, CY: Cyprus, NZ: New Zealand, PH: Philippines, UK: United Kingdom, CA: Canada.

#### 2.1.3 Environmental variables

In the selection of environmental variables, our primary focus was on processes within the terrestrial carbon cycle. The carbon cycle on land can be conceptualized as two flux exchange processes influenced by the climatic conditions (Fig. 4). The CO<sub>2</sub> in the atmosphere is fixed by vegetation photosynthesis and the carbon is released back into the atmosphere by respiration and disturbance processes (Beer et al., 2010; Pan et al., 2011). The carbon fluxes through these processes we considered as the land flux. Since Industrial Era, anthropogenic carbon from land use change (e.g., deforestation) and fossil fuels and cement become important parts of atmospheric CO<sub>2</sub> (Friedlingstein et al., 2010), which we considered as the anthropogenic flux. Meanwhile, the two processes are directly or indirectly driven by the climatic features (Sitch et al., 2015; Chen et al., 2021). Consequently, we explored the potential drivers of XCO<sub>2</sub> from the perspective of the carbon cycle at atmosphere-land interface. Multiple satellite products and reanalysis data from three aspects (i.e., land flux, anthropogenic flux and climatic impacts) were selected to consider their various effects on the XCO<sub>2</sub>.

Figure 4. Simplified illustration of the global carbon cycle on land (referring to IPCC

2023). Noting that the carbon cycle in the ocean was not considered in our study and we only focused on the fast exchange fluxes. The slow carbon exchanges (e.g., chemical weathering, volcanic emissions) which are generally assumed as relatively constant over the last few centuries (Sundquist, 1986), were not included here.

The key factors selected related to the land flux included gross primary productivity (GPP), enhanced vegetation index (EVI), land surface temperature (LST), vegetation continuous fields (VCF), and normalized difference snow index (NDSI). These products are all obtained from the Moderate Resolution Imaging Spectroradiometer (MODIS), which has been operated for over 20 years and produced various satellite products with fine spatial resolution and accuracy. The EVI and NDSI were converted to monthly data using the maximum value composite (MVC) method. The GPP and LST were converted to monthly data by the averaging method.

The rising anthropogenic activities have greatly influenced atmospheric CO<sub>2</sub> (Friedlingstein et al., 2022). In this study, five anthropogenic factors, including land use/cover change (LUCC), nighttime lights (NTL), and three trace gases (i.e., sulfur dioxide (SO<sub>2</sub>), nitrogen dioxide (NO<sub>2</sub>), and carbon monoxide (CO)) were selected. The LUCC was obtained from MODIS MCD12Q1 with a spatial resolution of 500 m. The monthly Suomi National Polar-orbiting Partnership-Visible Infrared Imaging Radiometer Suite (NPP-VIIRS) day/night band (DNB) NTL products (spatial resolution of 15 arc-second, ~500 m) were obtained from the Earth Observation Group (EOG) of the Colorado School of Mines. We also used the SO<sub>2</sub>, NO<sub>2</sub> and CO products from the TROPOspheric Monitoring Instrument (TROPOMI) onboard Sentinel-5 Precursor (S5P), a global air monitoring satellite for the Copernicus mission. The data were also converted to the same temporal resolution (i.e., monthly).

The selected climatic factors affecting XCO<sub>2</sub> were surface pressure (SP), 10 m wind speed (WS), precipitation flux (PRE), 2 m air temperature (Temp), and total evaporation (E). These data are from the reanalysis products (Hersbach et al., 2020) developed at the European Center for Medium Weather Forecasting (ECMWF, https://www.ecmwf.int/). The WS is calculated using the products of 10 m wind components of U and V. All data were converted to monthly time-series. The bilinear interpolation approach was employed both to fill gaps in the ancillary data and to convert the data at different spatial resolutions to 0.05° resolution. The data preprocessing was conducted on GEE, R and ArcGIS 10.3. Details of these products

**Table 2**. Ancillary variables selected in this study.

| Variables       | Spatial resolution | Temporal resolution | _                       |               |  |
|-----------------|--------------------|---------------------|-------------------------|---------------|--|
| GPP             | 500 m              | 8-day MOD17A2H      |                         | Land flux-    |  |
| EVI             | 1 km               | 16-day              | MOD13A2                 | related       |  |
| LST             | 1 km               | daily               | MOD11A1                 |               |  |
| VCF             | 250 m              | annual              | MOD44B                  |               |  |
| NDSI            | 500 m              | daily               | MOD10A1                 |               |  |
| LUCC            | 500 m              | annual              | MCD12Q1                 |               |  |
| NTL             | NTL 15 arc-second  |                     | VNP46A2                 | A             |  |
| $\mathrm{SO}_2$ |                    |                     | OFFL/L3_SO <sub>2</sub> | Anthropogenic |  |
| $NO_2$          | ~1 km              | daily               | OFFL/L3_NO <sub>2</sub> | flux-related  |  |
| CO              |                    |                     | OFFL/L3_CO              |               |  |
| SP              |                    |                     |                         |               |  |
| E               |                    |                     |                         |               |  |
| Wind-v          | 10.1               | .1.1                | EDACI 1                 | Climatic      |  |
| Wind-u          | ~10 km             | monthly             | ERA5-Land               | impacts       |  |
| Pre             |                    |                     |                         | _             |  |
| Temp            |                    |                     |                         |               |  |

# 2.2 Deep learning-based XCO2 reconstruction

Given the complexity temporal dependence and nonlinear relationship between  $XCO_2$  and the environmental variables, we selected the At-BiLSTM model to relate the  $XCO_2$  data with the 16 response variables affecting atmospheric  $CO_2$ , and further reconstruct the  $XCO_2$  data at a fine spatial resolution. The LSTM model is a variant of RNN that excels in modeling temporal sequences and capture long-range dependencies (Hochreiter and Schmidhuber, 1997; Graves et al., 2005), which is essential for understanding the seasonal variations of  $XCO_2$  and dynamic feedbacks between  $XCO_2$  and environmental drivers we selected. Each LSTM cell includes an input gate, a forget gate and an output gate. The forget gate  $f_t$  determines which information from the previous time step to forget (Eq. 1):

$$f_t = \sigma(W_f \cdot [h_{t-1}, x_t] + b_f) \tag{1}$$

where  $\sigma$ ,  $W_f$ ,  $[h_{t-1}, x_t]$ , and  $b_f$  denotes the sigmoid activation function, vectors of weights, concatenation of the hidden state at timestep t-1 and the current input, and the bias vector, respectively.

The input gate  $i_t$  governs the selective storage of the data in current time step, and the output from forget gate  $f_t$  and input gate  $i_t$  are combined in the cell state  $C_t$ 

(Eq. 2-3):

$$i_t = \sigma(W_i \cdot [h_{t-1}, x_t] + b_i) \tag{2}$$

$$C_t = f_t \cdot C_{t-1} + i_t \cdot \tanh(W_C \cdot [h_{t-1}, x_t] + b_C)$$
(3)

- where  $W_i$  and  $W_C$  denote the weight matrix for the input gate and the current cell
- state, respectively;  $b_i$  and  $b_c$  are the bias vector of the input gate and the current cell
- state, respectively;  $C_{t-1}$  and tanh represent the cell state at timestep t-1 and the
- activation function.
- Lastly, the output gate  $o_t$  controls the flow of information from the cell state to
- the next time step (Eq. 4).

$$o_t = \sigma(W_0 \cdot [h_{t-1}, x_t] + b_0) \tag{4}$$

- where  $W_o$  and  $b_o$  denotes the weight matrix and the bias vector of the output gate,
- respectively.
- These gate structures effectively manage the flow of information within the LSTM,
- enabling it to capture the temporal dependencies present in the data (Yuan et al., 2020;
- Wang et al., 2022). Bidirectional LSTM consists of two directional LSTM, in which the
- data flows forward and backward (Graves et al., 2013). The bidirectional structure was
- chosen to enhance the capability of LSTM by allowing the model to consider both past
- and future context in the time series, thereby providing a more comprehensive
- understanding of the underlying temporal dynamics.
- We also defined a multi-dimensional attention layer behind the BiLSTM to focus
- on more critical timesteps and give them higher weights (Bahdanau et al., 2016). This
- is particularly important when dealing with high-dimensional input data comprising
- multi-timestep variables, as it allows the model to assign different weights to different
- timesteps, thereby improving interpretability and predictive performance (Liu and Guo,
- 2019; Wang et al., 2024b). Based on this framework, the At-BiLSTM model offers a
- robust and flexible framework for linking XCO<sub>2</sub> with multiple environmental variables
- and reconstructing XCO<sub>2</sub> at a fine spatial resolution with improved accuracy and
- spatiotemporal consistency.
- The At-BiLSTM consists of one input layer, three Bidirectional LSTM (Bi-LSTM)
- layers, one attention layer, one dropout layer to prevent overfitting, and one fully
- connected layer (i.e., dense layer) for the final output. Each Bi-LSTM includes 512
- hidden units and tanh activation in both forward and backward directions. The attention
- mechanism learns a weight distribution over the time dimension using a Dense layer

with softmax activation, then multiplies these weights with the BiLSTM output to emphasize important time steps. The detailed deployment and output are provided in Table 3. The model was implemented using the TensorFlow and Keras deep learning APIs in Python. A time step of 3 was used, and the model was trained for 200 epochs with the mean squared error (MSE) as the loss function. A step-wise decay strategy was applied to the learning rate, where the rate was reduced by a factor of 10 every 50 epochs to enhance training stability and convergence. Prior to training, all input data were normalized using the mean and standard deviation of the dataset.

**Table 3.** Architecture of the At-BiLSTM model

| Layer Name   | Layer               | Parameters                        | Output size     |
|--------------|---------------------|-----------------------------------|-----------------|
| Bi-LSTM      | Input layer         | -                                 | 3×16            |
|              | Bi-LSTM1            | units = 512, activation = 'tanh'  | 3×1024          |
|              | Bi-LSTM2            | units = 512, activation = 'tanh'  | $3 \times 1024$ |
|              | Bi-LSTM3            | units = 512, activation = 'tanh'  | 3 ×1024         |
| Attention    | Attention Permute - |                                   | $1024 \times 3$ |
|              | Dense               | units = 3, activation = 'softmax' | $1024 \times 3$ |
|              | Permute             | -                                 | 3 ×1024         |
|              | Multiply            | -                                 | 3 ×1024         |
| Dropout      |                     | rate = 0.5                        |                 |
| Full-connect | Dense               | units = 1                         | 1               |

In this study, we adopted the sample-based cross-validation (CV) method to evaluate the model performance and the in-situ validation to assess the accuracy of reconstructed XCO<sub>2</sub> products. We also compared the reconstructed XCO<sub>2</sub> products with the original OCO XCO<sub>2</sub> products and the CAMS-EGG4 GHGs data. Four metrics, including coefficient of determination (R<sup>2</sup>), root mean squared error (RMSE), mean absolute error (MAE) and mean bias, were calculated as follow, to assess the model performance.

$$R^{2} = 1 - \frac{\sum_{i=1}^{n} (y_{i} - f_{i})^{2}}{\sum_{i=1}^{n} (y_{i} - \bar{y})^{2}}$$
 (5)

$$RMSE = \sqrt{\frac{\sum_{i=1}^{n} (y_i - f_i)^2}{n}}$$
 (6)

$$MAE = \frac{\sum_{i=1}^{n} |(f_i - y_i)|}{n}$$
 (7)

where n is the total number of data samples, and  $f_i$ ,  $y_i$  are the observed results and model-estimated results, respectively.

#### **3. Results**

# 3.1 Validation of the reconstructed XCO<sub>2</sub> product

#### 3.1.1 Model validation results

Given the distinct seasonal variation in XCO<sub>2</sub> concentrations, we conducted the sample-based CV to evaluate the model performance during different seasons (Fig. 5). The model demonstrated high accuracy across all seasons, with R<sup>2</sup> values exceeding 0.81, MAE less than 0.73 ppm, and RMSE less than 1.09 ppm. The model performed better in spring and summer, as indicated by the densest cluster of points being closest to the 1:1 line. Conversely, the model performed worst in winter, when photosynthesis is weakest, leading to greater estimation deviation. These variations are likely influenced by the ecosystem CO<sub>2</sub> exchange during different seasons. Overall, the model effectively captured the seasonal variation of XCO<sub>2</sub> and provided unbiased XCO<sub>2</sub> estimations.

**Figure 5.** Density scatterplots of sample-based CV results during different seasons. The proportion of the number of points is represented as the color of the points. The black

dashed lines and grey solid lines denote the linear regression fitted lines and the 1:1 line, respectively. The  $R^2$ , RMSE (ppm), MAE (ppm), and mean bias (ppm) are provided.

We further validated the model performance across different continents. Table 4 presents the validation results for six continents. The model performance varied across continents. Notably, the model achieved the highest accuracy in Africa and Europe, with R<sup>2</sup> of 0.80 and 0.81, and RMSE values of 1.02 and 1.14 ppm, respectively. In contrast, the model demonstrated relatively low accuracy in Oceania and South America, both located in the southern hemisphere. Despite this, the RMSE of the model in these continents were 1.22 and 0.66 ppm, respectively, indicating that the model maintained acceptable estimation accuracy in these regions.

**Table 4**. Model performance in different continents.

|               | $\mathbb{R}^2$ | RMSE (ppm) | MAE (ppm) | Mean bias (ppm) |
|---------------|----------------|------------|-----------|-----------------|
| Africa        | 0.80           | 1.02       | 0.70      | -0.009          |
| Asia          | 0.73           | 1.27       | 0.85      | 0.002           |
| Europe        | 0.81           | 1.14       | 0.77      | -0.030          |
| North America | 0.73           | 1.26       | 0.83      | -0.020          |
| South America | 0.59           | 1.22       | 0.86      | -0.012          |
| Oceania       | 0.67           | 0.66       | 0.4       | 0.051           |

## 3.1.2 In situ validation results

The TCCON in situ XCO<sub>2</sub> data were adopted for validating the accuracy of the reconstructed XCO<sub>2</sub> over the globe. The validation results for our reconstructed XCO<sub>2</sub> and the origin OCO-2/3 XCO<sub>2</sub> are displayed in Fig. 6. The two XCO<sub>2</sub> data showed similar precision with the  $R^2$  value of 0.91 and 0.92, respectively (Fig. 6c-d). While the reconstructed XCO<sub>2</sub> greatly increases the data coverage with the validation sample increasing from 578 to 1432. Meanwhile, the reconstructed XCO<sub>2</sub> has a smaller RMSE and MAE with values of 1.58 and 1.22 ppm, respectively, compared with the OCO XCO<sub>2</sub>. These results indicate that the reconstructed XCO<sub>2</sub> had a closer agreement with TCCON XCO<sub>2</sub>. We also displayed the mean bias of OCO and reconstructed XCO<sub>2</sub> in each TCCON site (Fig. 6a-b). As shown in Fig. 6a, the OCO-2/3 observation tend to overestimate the XCO<sub>2</sub>, while the reconstructed XCO<sub>2</sub> could amend the underestimation of OCO XCO<sub>2</sub>. Over 68% of the validation sites of reconstructed XCO<sub>2</sub> had a mean bias less between  $\pm$  0.4 ppm. Given the orbital constraints of the ISS (Eldering et al.,

2019), OCO-3 measurements were restricted to latitudes below  $\pm$  52°. Consequently, substantial missing values of OCO XCO<sub>2</sub> data were shown around 50°N, introducing a potential bias. In contrast, the reconstructed XCO<sub>2</sub> effectively solves this problem and demonstrates markedly enhanced performance.

**Figure 6.** The mean bias of the (a) OCO observed  $XCO_2$ , and (b) reconstructed  $XCO_2$  against global TCCON  $XCO_2$ ; (c) density scatterplots of the validation results for OCO observed  $XCO_2$ , and (d) reconstructed  $XCO_2$  against the TCCON  $XCO_2$ . The proportion of the number of points is represented as the color of the points. The number of samples n, linear regression relation,  $R^2$ , RMSE (ppm), MAE (ppm), and mean bias are provided.

Fig. 7 shows the individual in situ validation results of the reconstructed XCO<sub>2</sub> against TCCON site in different continents (except Antarctica). The sample numbers are varying in different sites due to the observation constraints, while the validation results from all sites showed satisfying performance. The R<sup>2</sup> for all sites are over 0.88 and the MAE are less than 1.46 ppm. The reconstructed XCO<sub>2</sub> data performs the best in sites lauder03 and karlsruhe01, which located in North America and Europe, respectively. While the reconstructed XCO<sub>2</sub> performed worst in saga01 which located in Asia, potentially due to the high CO<sub>2</sub> concentrations in these regions. Overall, the reconstructed XCO<sub>2</sub> showed high consistency with the in situ XCO<sub>2</sub> observation in different regions over the globe.

**Figure 7.** Scatterplots of the TCCON in situ validation results of the reconstructed XCO<sub>2</sub> on different TCCON sites over the globe.

To assess the performance of our reconstructed XCO<sub>2</sub> in temporal analysis, we compared the time series for monthly OCO-2/3, reconstructed and TCCON XCO<sub>2</sub> data from December 2014 to December 2021. As depicted in Fig. 8, the reconstructed XCO<sub>2</sub> exhibits similar temporal patterns compared to the TCCON data, with the mean RMSE and MAE of 1.47 and 1.07 ppm. While the OCO-2/3 XCO<sub>2</sub> exhibits some overestimation for high values and underestimation for low values compared with TCCON data. In contrast, the reconstructed XCO<sub>2</sub> provided more stable estimate results.

**Figure 8.** Comparison of the temporal variation of XCO<sub>2</sub> data from OCO-2/3 (blue dots), TCCON (green dots), and the reconstructed products (yellow dots).

## 3.2 Spatiotemporal pattern of global XCO<sub>2</sub>

The global distribution of annual mean XCO<sub>2</sub> concentration from 2015 to 2021 is illustrated in Fig. 9. The results reveal pronounced spatial heterogeneity in XCO<sub>2</sub> concentrations, characterized by a marked hemispheric asymmetry. Specifically, the Northern Hemisphere exhibited systematically elevated XCO<sub>2</sub> levels compared to the Southern Hemisphere, consistent with latitudinal gradients driven by anthropogenic emission patterns and atmospheric transport dynamics. Regionally, North America, East Asia, Central Africa, and northwest of Southern America were identified as persistent hotspots of enhanced XCO<sub>2</sub>. The high concentrations of XCO<sub>2</sub> in North America and East Asia stem primarily from the fossil fuel emission from energy production and transportation sectors. Whereas the tropical regions (i.e., Central Africa and South America) are influenced by coupled biomass burning and land-use changes.

**Figure 9.** The global spatial distribution of reconstructed annual mean XCO<sub>2</sub> concentration from 2015 to 2021.

We also provided the annual OCO-2 XCO<sub>2</sub> data from 2015 to 2019 and OCO-3 XCO<sub>2</sub> data from 2020 to 2021 in Fig. 10. Spatially, our reconstructed XCO<sub>2</sub> dataset (Fig. 9) demonstrates robust consistency with satellite observations, particularly in midlatitude industrialized regions where both datasets capture emission hotspots. Notably, OCO-3 exhibits denser observational sampling due to its improved spatial coverage and

swath width compared to OCO-2's narrow tracks. However, persistent data gaps remain prevalent in both two satellite products after annual aggregating. These spatial coverage limitations hinder fine-scale global analysis, particularly in assessing localized emission sources and regional scale carbon flux.

**Figure 10.** The global spatial distribution of annual mean OCO-2/OCO-3 XCO<sub>2</sub> concentration from 2015 to 2021.

Fig. 11 presents the spatial distribution of the 7-year (2015-2021) averaged XCO<sub>2</sub> concentration and trend over the globe. The average XCO<sub>2</sub> concentration from 2015 to 2021 was 406.90 ± 0.80 ppm worldwide. The highest concentration of XCO<sub>2</sub> mainly occurs in the northern low-to-mid-latitudes (10°N-45°N). More frequent human activities and carbon emissions contributed to higher atmospheric CO<sub>2</sub> concentrations in the Northern Hemisphere. In contrast, the lowest XCO<sub>2</sub> concentration was 404.02 ppm, occurring in the Southern Hemisphere where 81% of the area is ocean. The oceans act as a vital carbon sink and absorb most atmospheric CO<sub>2</sub>. For the continent scale, the XCO<sub>2</sub> concentrations showed a slight variation (±1 ppm) between different continents. The largest XCO<sub>2</sub> were mainly occurred in Asia and North America over years, while the lowest XCO<sub>2</sub> concentration all presented in Oceania (Table 4). In terms of temporal trend, the atmospheric CO<sub>2</sub> exhibited a distinct increasing trend over time, with the mean growth rate of 2.32 ppm yr<sup>-1</sup>. The large growth rate meanly occurs in the northern

low latitudes (0°N-30°N), especially the Middle East and North Africa (growth rate over 2.5 ppm yr<sup>-1</sup>). Globally, the XCO<sub>2</sub> increased by 14.16 ppm over seven years (Table 4), especially in 2021, with increased values of up to 3 ppm. This result is consistent with the Global Carbon Budget 2022 (Friedlingstein et al., 2022), which reported that the global average atmospheric CO<sub>2</sub> increased sharply in 2021 and reached 414.71 ppm.

**Figure 11.** The global spatial distribution of (a) reconstructed 7-year averaged XCO<sub>2</sub> concentration, and (b) its trend from 2015 to 2021 (ppm yr<sup>-1</sup> denotes parts per million per year).

**Table 4.** The reconstructed XCO<sub>2</sub> concentrations at different continents from 2015 to 2021.

| Continents    | XCO <sub>2</sub> concentrations (ppm) |        |        |        |        |        |        |          |
|---------------|---------------------------------------|--------|--------|--------|--------|--------|--------|----------|
|               | 2015                                  | 2016   | 2017   | 2018   | 2019   | 2020   | 2021   | Increase |
| Africa        | 399.26                                | 402.66 | 404.98 | 406.71 | 409.26 | 411.13 | 414.11 | 14.85    |
| Asia          | 399.57                                | 403.03 | 405.80 | 407.37 | 409.68 | 411.39 | 414.38 | 14.81    |
| Europe        | 399.55                                | 402.88 | 405.77 | 406.96 | 409.48 | 411.30 | 414.17 | 14.62    |
| North America | 399.60                                | 402.95 | 405.76 | 407.32 | 409.70 | 411.61 | 414.28 | 14.68    |
| South America | 398.94                                | 401.96 | 404.27 | 406.17 | 408.78 | 410.47 | 413.57 | 14.63    |
| Oceania       | 398.03                                | 401.04 | 403.31 | 405.53 | 408.13 | 409.82 | 412.55 | 14.52    |
| Global        | 399.84                                | 401.56 | 405.16 | 407.50 | 409.21 | 411.07 | 414.00 | 14.16    |

# 3.3 The distribution of XCO<sub>2</sub> anomaly

To better explore the dynamics of global carbon change, we further calculated the XCO<sub>2</sub> anomalies based on the full-coverage XCO<sub>2</sub> products and presented their global distributions from 2015 to 2021 (Fig. 12). The XCO<sub>2</sub> anomalies were calculated by the statistical filtering method, that is, subtracting the global median XCO<sub>2</sub> value from the global XCO<sub>2</sub> distribution (Hakkarainen et al., 2016). The spatial pattern of XCO<sub>2</sub> anomalies were relatively consistent over seven years with no significant variations. From the global perspective, high XCO<sub>2</sub> anomalies mainly occurred in the Northern Hemisphere. East Asia has the largest XCO<sub>2</sub> anomalies with values ranging from 2 to 3 ppm, such as the east part of China. The Middle East, North Africa and the southern part of Northern America also experienced high XCO2 anomalies. Nevertheless, negative XCO<sub>2</sub> anomalies were also identified in the Northern Hemisphere, specifically in regions such as Tibet in China, eastern Canada, and southern Russia. Most negative XCO<sub>2</sub> anomalies were observed in the Southern Hemisphere, which behaves as a carbon sink. However, some positive XCO<sub>2</sub> anomalies are also observed in the tropical regions (e.g., Amazonia), which indicates the Amazonia has changed into a carbon source due to the deforestation and fire occurrence in recent years (Hubau et al., 2020; Gatti et al., 2021).

Figure 12. The global spatial distribution of annual XCO<sub>2</sub> anomaly from 2015 to 2021.

Fig. 13 illustrates the detailed spatial distribution of XCO<sub>2</sub> concentrations and anomalies over six regions with high XCO<sub>2</sub> retrievals in 2020. High concentrations of XCO<sub>2</sub> were typically associated with energy-intensive heavy industrial activities, such as Toa Oil Keihin Refinery Factory located in Kawasaki City, Japan (Fig. 13f), and the Shippingport Industrial Park in Pennsylvania, United States (Fig. 13a). Moreover, certain metropolitan transport hubs also exhibited elevated CO<sub>2</sub> anomalies attributable to dense populations and intensive activities. Examples included Shanghai Station in China (Fig.13e) and John F. Kennedy International Airport in New York, USA (Fig. 13b). Attention has also been drawn to natural sources of emissions. Driven by the significant impact of agricultural mechanization and agro-industrial activities on cropland (Lin and Xu, 2018), the XCO<sub>2</sub> anomalies also occurred in the agricultural areas northwestern Jiangsu, China (Fig. 13d). Additionally, we also observed the high XCO<sub>2</sub> anomalies in Amazonia forest in Colombia, which have been suffered from deforestation (Gatti et al., 2023). In conclusion, our products could successfully capture the XCO<sub>2</sub> anomalies from different sources over the globe.

**Figure 13.** Examples of XCO<sub>2</sub> hotspots in six regions for 2020 detected using the reconstructed products. The subplots present the spatial distribution of XCO<sub>2</sub> concentrations, anomalies (the red panels), and the emission sources (the true color images from Google Earth), respectively. The global map in the middle presents the land use and land cover types over the globe.

#### 4. Discussion

### 4.1 Comparison with previous studies

To validate the effectiveness of our model and resulting XCO<sub>2</sub> products, we compared our results with current studies which focuses on global XCO<sub>2</sub> reconstruction (Table 5). As for the in-situ validation, most existing studies report high accuracy with almost all R<sup>2</sup> over 0.9, RMSE less than 2 ppm. Regarding spatial resolution, the various products differ substantially, ranging from 1° down to 0.01°. It should be noted that increasing spatial resolution tends to compromise the accuracy of XCO<sub>2</sub> retrievals. However, our XCO<sub>2</sub> product achieves an optimal balance between spatial detail and measurement precision, exhibiting both high spatial resolution (0.05°) and robust accuracy (R<sup>2</sup>=0.91, RMSE =1.54 ppm) in comprehensive evaluations.

**Table 5.** Comparison between current studies focusing on global XCO<sub>2</sub> reconstruction

524525

536537

| Model                  | Spatial resolution | In-situ validation<br>(with TCCON) |                         |       | Reference           |
|------------------------|--------------------|------------------------------------|-------------------------|-------|---------------------|
|                        |                    | R <sup>2</sup>                     | R <sup>2</sup> RMSE MAE |       | -                   |
|                        |                    |                                    | (ppm)                   | (ppm) |                     |
| Attentional-based LSTM | 0.05°              | 0.91                               | 1.54                    | 1.22  | Our study           |
| Deep forest            | 0.1°               | 0.96                               | 1.01                    | -     | Zhang et al. (2023) |
| S-STDCT                | $0.25^{\circ}$     | 0.95                               | 1.18                    | -     | Wang et al. (2023)  |
| Spatiotemporal kriging | 1°                 | 0.97                               | 1.13                    | 0.88  | Sheng et al. (2022) |
| MLE & OI               | 0.5°               | 0.92                               | 2.62                    | 1.53  | Jin et al. (2022)   |
| ERT                    | 0.01°              | 0.83                               | 1.79                    | -     | Li et al. (2022)    |

\*S-STDCT: Self-supervised spatiotemporal discrete cosine transform; MLE & OI: maximum likelihood estimation method and optimal interpolation; ERT: Extremely randomized trees

To evaluate the advancement of our XCO<sub>2</sub> product, we compared it with original OCO-2 observations and publicly available global XCO<sub>2</sub> datasets (Wang et al., 2023; Sheng et al., 2022; Zhang et al., 2023) across four regions: North America, Europe with northern Africa, Asia, and Oceania (Fig. 14) in January 2015. Despite monthly aggregation, OCO-2 data exhibit persistent spatial discontinuities, limiting the capacity to analyze monthly XCO<sub>2</sub> variability at regional and national scales. Existing XCO<sub>2</sub> products (spatial resolution of 0.25°, 1°, and 0.1°, respectively) broadly reproduce large-scale XCO<sub>2</sub> patterns but fail to resolve fine-scale heterogeneity. In comparison, our reconstructed XCO<sub>2</sub>, with the highest spatial resolution, provides a more detailed and accurate representation of the regional XCO<sub>2</sub> patterns. For example, lower XCO<sub>2</sub> concentrations are clearly identified in eastern Canada (The first row of Fig.14) and Papua New Guinea (The fourth row of Fig. 14), regions characterized by dense forest cover. This correspondence highlights the substantial carbon sink potential of these forested areas. Our high-resolution product better identifies the CO<sub>2</sub> heterogeneity associated with different land cover types, whereas the coarse-resolution products smooth these signals. This limitation primarily stems from the neglect of highresolution land cover dynamics and dependence on coarse-resolution assimilated/reanalysis datasets (e.g., CAMS XCO2, CarbonTracker), resulting in oversmoothed spatial patterns that obscure satellite-derived high-resolution signals. Unlike assimilation-dependent approaches, our method avoids XCO<sub>2</sub> reanalysis inputs, preserving satellite-scale fidelity through high-resolution environmental variables modeling while maintaining precision.

**Figure 14.** Comparison between the OCO-2 XCO<sub>2</sub> data, accessible XCO<sub>2</sub> products from Wang et al. (2023), Sheng et al. (2022), Zhang et al. (2023), and our reconstructed XCO<sub>2</sub> data in four regions, using the products of January of 2015 as an example.

# 4.2 Limitations and future improvements

Though our XCO<sub>2</sub> products achieved full spatial coverage and high accuracy, however, there are still several limitations need further improvement. In terms of the satellite data, OCO-2 and OCO-3 provide different spatiotemporal coverages. Analyzing OCO-2 and OCO-3 data simultaneously may introduce several uncertainties due to these differences. However, OCO-3 has a similar sensor and inherits the retrieval algorithms of OCO-2. According to Taylor et al. (2023), the mean differences between OCO-3 and OCO-2 are around 0.2 ppm over land. Therefore, we suppose that the discrepancies between their datasets are minimal, and the combined analysis of data from these two satellites will have a negligible impact on our results.

Additionally, though our model integrates multiple environmental variables associated with surface carbon flux variations, it does not account for vertical atmospheric transport. As XCO<sub>2</sub> represents the column-averaged CO<sub>2</sub> concentration, vertical redistribution of CO<sub>2</sub> through atmospheric transport (e.g., mixing, convection) can alter the relationship between surface carbon fluxes and column concentrations (Shirai et al., 2012). The absence of such vertical transport indicators may reduce the model's accuracy in regions or periods with strong vertical mixing. Future efforts will

incorporate vertical transport-related variables, such as planetary boundary layer height, vertical wind components, and other reanalysis-derived indicators, to better represent the atmospheric processes that influence the column-averaged CO<sub>2</sub> signal.

Moreover, while OCO missions currently provide some of the most accurate carbon satellite-based XCO<sub>2</sub> retrievals, they still encounter some retrieval errors and data gaps driven by algorithmic limitations and variable meteorological conditions. A critical research frontier is the refinement of XCO<sub>2</sub> retrieval algorithms to mitigate systematic biases in high-aerosol-load regions (e.g., industrial regions and biomass-burning plumes). Additionally, next-generation hyperspectral satellites, such as the upcoming CO2M (Copernicus Anthropogenic CO<sub>2</sub> Monitoring Mission) with 2×2 km<sup>2</sup> resolution and GeoCarb (Geostationary Carbon Observatory) offering hourly monitoring, will enhance spatial-temporal coverage and reduce cloud-induced data gaps (Reuter et al., 2025).

# 5. Data availability

The XCO<sub>2</sub> dataset produced in this paper is available at https://doi.org/10.5281/zenodo.12706142 (Wang et al., 2024a). It includes monthly global XCO<sub>2</sub> data at 0.05° resolution, covering the period from December 2014 to December 2021. The dataset is archived in netCDF4 format, with units in parts per million (ppm).

## 6. Conclusion

As a major driver of global warming, the monitoring of CO<sub>2</sub> changes, especially anthropogenic CO<sub>2</sub> emissions, is of critical importance. The launch of carbon satellites offers a significant advancement for CO<sub>2</sub> monitoring. However, the limited spatial coverage of satellite observations constrains the utility of XCO<sub>2</sub> data. While current XCO<sub>2</sub> products exhibit relatively high validation accuracy, their coarse spatial resolution remains inadequate for applications such as regional- or county-level emission monitoring, as well as for the detection and inversion of large emission sources. To address these issues, we reconstructed a global full-coverage XCO<sub>2</sub> product at a fine spatial resolution of 0.05° and temporal resolution of 1 month from 2015 to 2021. The advanced deep learning method was adopted to model time-series XCO<sub>2</sub> and incorporate terrestrial flux, anthropogenic flux and climatic impacts into the

- parameterization process. Through comprehensive evaluations, including cross-validation, in-situ validation, spatial distribution assessment and comparison with other XCO<sub>2</sub> products, our reconstructed XCO<sub>2</sub> products demonstrates significant improvements in both accuracy and spatial resolution. The main conclusions and contributions are as following:
  - (1) The advanced At-BiLSTM model could successfully established the nonlinear relationship between satellite-derived XCO<sub>2</sub> and a set of key environmental variables. And the reconstructed XCO<sub>2</sub> based on our model shows relatively good agreement with TCCON XCO<sub>2</sub>, with R<sup>2</sup>, RMSE, and MAE values of 0.91, 1.58 ppm, and 1.22 ppm, respectively.
  - (2) The reconstructed XCO<sub>2</sub> product overcomes the extensive data gaps typically caused by narrow satellite swaths and retrieval interference from clouds and aerosols, achieving complete global coverage. Moreover, relative to existing publicly available full-coverage XCO<sub>2</sub> datasets, our product offers the finest spatial resolution (0.05°) while maintaining comparable accuracy.
  - (3) Our method avoids coarse XCO<sub>2</sub> reanalysis inputs, preserving satellite-scale fidelity through high-resolution environmental variables modeling. Consequently, the products enable enhanced ability in identifying regional- and county-level XCO<sub>2</sub> hotpots, carbon emissions and fragmented carbon sinks, providing a robust basis for targeted global carbon governance policies.

#### Acknowledgements

620

- The authors would like to express their gratitude to the NASA Goddard Earth Science
- Data and Information Services Center for providing the OCO-2/3 XCO<sub>2</sub> products
- (https://disc.gsfc.nasa.gov/), the NASA Land Processes Distributed Active Archive
- Center (LP DAAC) for providing MODIS data. Our gratitude also goes to the Earth
- Observation Group (EOG) of the Colorado School of Mines for supplying the NPP-
- VIIRS NTL products, the European Space Agency (ESA) for providing the TROPOMI
- data, the Copernicus Climate Data Store for providing the ERA5 reanalysis data and
- CAMS-EGG4 XCO<sub>2</sub> data, and the Total Carbon Column Observing Network for their
- dedication in providing the in situ XCO<sub>2</sub> observations.

#### **Author contributions**

ZW developed the overall workflow, processed the data and wrote the manuscript. CZ

- and BH revised the manuscript. KS, YS, and XC compiled the data. SC conceptualized
- and revised the manuscript. PA and QZ supervised this study. All the authors
- contributed to the study.

#### 625 **Competing interests**

The contact author has declared that none of the authors has any competing interests.

#### **Financial support** 627

- This work was supported by the Key R&D Program of Zhejiang (2022C03078) and 628
- National Natural Science Foundation of China (32241036).

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
