# Peer review of "A full-coverage satellite-based global atmospheric CO2 dataset at 0.05°"

_Earth System Science Data, 2024_

## Author Comment (AC1)

**Response to Anonymous Referee #1:**

This manuscript presents the development of a global dataset of column-averaged dry-air mole fraction of $CO_2$ ($XCO_2$) at high resolution (0.05°) using multi satellite products, and an improved deep learning model. They further evaluate new datasets using the measurements from the TCCON network. While the study shows promise, there are significant concerns regarding methodological transparency, and clarity of explicit demonstration of advantages over existing satellite products. Addressing these issues will greatly enhance the manuscript's impact and originality.

**Response:** We sincerely appreciate your thorough review of our manuscript and the valuable, constructive feedback. In response, we have expanded our discussion of the methods, and the advantages of our products over existing $XCO_2$ data. Our point-by-point responses to your comments are provided below.

**Q1.** (1) The manuscript lacks in describing the methodological details of the improved deep learning model. The authors should clearly outline the specific innovations or modifications that lead to improved accuracy.

**Response:** Thanks for the suggestion. Given the complex temporal dependencies and nonlinear relationships between atmospheric $XCO_2$ and a wide range of environmental variables, we selected the Attention-based Bidirectional Long Short-Term Memory (At-BiLSTM) model for this study. This choice is motivated by several key considerations:

Firstly, LSTM networks are well-suited for modeling temporal sequences and capturing long-range dependencies, which is essential for understanding the seasonal variations of $XCO_2$ and dynamic feedbacks between $XCO_2$ and environmental drivers such as temperature, vegetation activity, and surface pressure. The bidirectional structure enhances this capability by allowing the model to consider both past and future context in the time series, thereby providing a more comprehensive representation of the underlying temporal dynamics.

Secondly, the incorporation of the attention mechanism enables the model to dynamically focus on the most critical time steps when making predictions. This is particularly important when dealing with high-dimensional input data comprising

multi-timestep variables, as it allows the model to assign different weights to different input features, thereby improving interpretability and predictive performance.

Finally, the At-BiLSTM model's ability to capture nonlinear relationships is crucial in the context of atmospheric $CO_2$ modeling, where interactions between variables are complex and nonlinear. By leveraging the strengths of deep learning, the model can learn intricate patterns from the multi-source data that are difficult to capture with traditional statistical or linear models.

Therefore, we chose At-BiLSTM model as a robust and flexible framework to reconstructing $XCO_2$ at fine spatial resolution with improved accuracy and spatiotemporal consistency.

We have included the necessary clarifications in **2.2 Deep learning-based XCO₂ reconstruction**:

"The LSTM model is a variant of RNN that excels in modeling temporal sequences and capture long-range dependencies (Hochreiter and Schmidhuber, 1997; Graves et al., 2005), which is essential for understanding the seasonal variations of $XCO_2$ and dynamic feedbacks between $XCO_2$ and environmental drivers we selected. Each LSTM cell includes an input gate, a forget gate and an output gate. The forget gate $f_t$ determines which information from the previous time step to forget (Eq. 1):

$$f_t = \sigma(W_f \cdot [h_{t-1}, x_t] + b_f) \tag{1}$$

where $\sigma$, $W_f$, $[h_{t-1}, x_t]$, and $b_f$ denotes the sigmoid activation function, vectors of weights, concatenation of the hidden state at timestep $t$-$1$ and the current input, and the bias vector, respectively.

The input gate $i_t$ governs the selective storage of the data in current time step, and the output from forget gate $f_t$ and input gate $i_t$ are combined in the cell state $C_t$ (Eq. 2-3):

$$i_t = \sigma(W_i \cdot [h_{t-1}, x_t] + b_i) \tag{2}$$

$$C_t = f_t \cdot C_{t-1} + i_t \cdot tanh(W_C \cdot [h_{t-1}, x_t] + b_C) \tag{3}$$

where $W_i$ and $W_C$ denote the weight matrix for the input gate and the current cell state, respectively; $b_i$ and $b_c$ are the bias vector of the input gate and the current cell state, respectively; $C_{t-1}$ and $tanh$ represent the cell state at timestep $t$-$1$ and the activation function.

Lastly, the output gate $o_t$ controls the flow of information from the cell state to the next time step.

$$o_t = \sigma(W_o \cdot [h_{t-1}, x_t] + b_o) \qquad (4)$$

where $W_o$ and $b_o$ denotes the weight matrix and the bias vector of the output gate, respectively.

These gate structures effectively manage the flow of information within the LSTM, enabling it to capture the temporal dependencies present in the data (Yuan et al., 2020; Su et al., 2021). Bidirectional LSTM consists of two directional LSTM, in which the data flows forward and backward (Graves et al., 2013). The bidirectional structure was chosen to enhance the capability of LSTM by allowing the model to consider both past and future context in the time series, thereby providing a more comprehensive understanding of the underlying temporal dynamics.

We also defined a multi-dimensional attention layer behind the BiLSTM to focus on more critical timesteps and give them higher weights (Bahdanau et al., 2016). This is particularly important when dealing with high-dimensional input data comprising multi-timestep variables, as it allows the model to assign different weights to different timesteps, thereby improving interpretability and predictive performance (Liu and Guo, 2019). Based on this framework, the At-BiLSTM model offers a robust and flexible framework for linking $XCO_2$ with multiple environmental variables and reconstructing $XCO_2$ at fine spatial resolution with improved accuracy and spatiotemporal consistency."

And we have also added the detailed deployment and output of this deep learning model as follows:

"The At-BiLSTM consists of one input layer, three Bidirectional LSTM (Bi-LSTM) layers, one attention layer, one dropout layer to prevent overfitting, and one fully connected layer (i.e., dense layer) for the final output. Each Bi-LSTM includes 512 hidden units and tanh activation in both forward and backward directions. The attention mechanism learns a weight distribution over the time dimension using a Dense layer with softmax activation, then multiplies these weights with the BiLSTM output to emphasize important time steps. The detailed deployment and output are provided in Table 3. The model was implemented using the TensorFlow and Keras deep learning APIs in Python. A time step of 3 was used, and the model was trained for 200 epochs with the mean squared error (MSE) as the loss function. A step-wise decay strategy was applied to the learning rate, where the rate was reduced by a factor of 10 every 50 epochs to enhance training stability and convergence. Prior to training, all input data were normalized using the mean and standard deviation of the dataset."

**Table 3.** Architecture of the At-BiLSTM model

| Layer Name | Layer | Parameters | Output size |
|---|---|---|---|
| Bi-LSTM | Input layer | | 3×16 |
| | Bi-LSTM1 | units = 512, activation = 'tanh' | 3×1024 |
| | Bi-LSTM2 | units = 512, activation = 'tanh' | 3 × 1024 |
| | Bi-LSTM3 | units = 512, activation = 'tanh' | 3 ×1024 |
| Attention | Permute | - | 1024×3 |
| | Dense | units = 3, activation = 'softmax' | 1024×3 |
| | Permute | - | 3 ×1024 |
| | Multiply | - | 3 ×1024 |
| Dropout | | rate = 0.5 | |
| Full-connect | Dense | units = 1 | 1 |

(2) Additionally, since the formation and distribution of $XCO_2$ are influenced by atmospheric transport processes across multiple vertical layers (and not solely by surface fluxes), it is important that the manuscript explains how these vertical transport processes are incorporated into the model. If these processes are not accounted for, this limitation should be explicitly acknowledged.

**Response:** Thank you for raising this important point. In this study, we estimated column-averaged $CO_2$ ($XCO_2$), and as such, vertical transport processes were not explicitly incorporated into the modeling framework. However, we acknowledge that vertical redistribution of $CO_2$ through atmospheric transport processes (e.g., mixing and convection) can significantly influence the spatiotemporal patterns of $XCO_2$, particularly by altering the linkage between surface fluxes and column concentrations. The absence of vertical transport indicators in our model may limit its accuracy, especially in regions or periods characterized by strong vertical mixing. We have included a discussion of this limitation and have highlighted it as a key area for future model enhancement in **4.2 Limitations and future improvements**:

"Additionally, though our model integrates multiple environmental variables associated with surface carbon flux variations, it does not account for vertical atmospheric transport. As $XCO_2$ represents the column-averaged $CO_2$ concentration, vertical redistribution of $CO_2$ through atmospheric transport (e.g., mixing, convection) can alter the relationship between surface carbon fluxes and column concentrations.

The absence of such vertical transport indicators may reduce the model's accuracy in regions or periods with strong vertical mixing. Future efforts will incorporate vertical transport-related variables, such as planetary boundary layer height, vertical wind components, and other reanalysis-derived indicators, to better represent the atmospheric processes that influence the column-averaged $CO_2$ signal."

**Q2.** Although the authors present a new global $XCO_2$ product at 0.05° resolution, the distinct novelty and advantages over existing datasets remain unclear. The study should explicitly state how the analyzed information significantly differs from or improves upon existing satellite data. While validation against TCCON is good, the authors should explicitly compare these results with those from existing datasets to clearly demonstrate accuracy improvements.

**Response:** Thanks for the constructive suggestion. To validate the effectiveness of our model and resulting $XCO_2$ products, we firstly compared our results with current studies which focuses on global $XCO_2$ reconstruction. Afterwards, we compared our products with original OCO-2 observations and three publicly available global $XCO_2$ datasets to evaluate the advancement of our $XCO_2$ product. The comparison results have added in **4.1 Comparison with previous studies** as follow:

"To validate the effectiveness of our model and resulting $XCO_2$ products, we compared our results with current studies which focuses on global $XCO_2$ reconstruction (Table 5). As for the in-situ validation, most existing studies report high accuracy with almost all $R^2$ over 0.9, RMSE less than 2 ppm. Regarding spatial resolution, the various products differ substantially, ranging from 1° down to 0.01°. It should be noted that increasing spatial resolution tends to compromise the accuracy of $XCO_2$ retrievals. However, our $XCO_2$ product achieves an optimal balance between spatial detail and measurement precision, exhibiting both high spatial resolution (0.05°) and robust accuracy ($R^2$=0.91, RMSE =1.54 ppm) in comprehensive evaluations.

**Table 5.** Comparison between current studies focusing on global $XCO_2$ reconstruction

| Model | Spatial resolution | In-situ validation (with TCCON) | | | Reference |
|---|---|---|---|---|---|
| | | $R^2$ | RMSE (ppm) | MAE (ppm) | |
| Attentional-based LSTM | 0.05° | 0.91 | 1.54 | 1.22 | Our study |
| Deep forest | 0.1° | 0.96 | 1.01 | - | Zhang et al. (2023) |
| S-STDCT | 0.25° | 0.95 | 1.18 | - | Wang et al. (2023) |
| Spatiotemporal kriging | 1° | 0.97 | 1.13 | 0.88 | Sheng et al. (2022) |
| MLE & OI | 0.5° | 0.92 | 2.62 | 1.53 | Jin et al. (2022) |
| ERT | 0.01° | 0.83 | 1.79 | - | Li et al. (2022) |

*S-STDCT: Self-supervised spatiotemporal discrete cosine transform, MLE & OI: maximum likelihood estimation method and optimal interpolation; ERT: Extremely randomized trees

To evaluate the advancement of our $XCO_2$ product, we compared it with original OCO-2 observations and publicly available global $XCO_2$ datasets (Wang et al., 2023; Sheng et al., 2022; Zhang et al., 2023) across four regions: North America, Europe with northern Africa, Asia, and Oceania (Fig. 13) in January 2015. Despite monthly aggregation, OCO-2 data exhibit persistent spatial discontinuities, limiting the capacity to analyze monthly $XCO_2$ variability at regional and national scales. Existing $XCO_2$ products (spatial resolution of 0.25°, 1°, and 0.1°, respectively) broadly reproduce large-scale $XCO_2$ patterns but fail to resolve fine-scale heterogeneity. In comparison, our reconstructed $XCO_2$, with the highest spatial resolution, provides a more detailed and accurate representation of the regional $XCO_2$ patterns. For example, lower $XCO_2$ concentrations are clearly identified in eastern Canada (The first row of Fig.13) and Papua New Guinea (The fourth row of Fig. 13), regions characterized by dense forest cover. This correspondence highlights the substantial carbon sink potential of these forested areas. Our high-resolution product better identifies the $CO_2$ heterogeneity associated with different land cover types, whereas the coarse-resolution products smooth these signals. This limitation primarily stems from the neglect of high-resolution land cover dynamics and dependence on coarse-resolution assimilated/reanalysis datasets (e.g., CAMS $XCO_2$, CarbonTracker), resulting in oversmoothed spatial patterns that obscure satellite-derived high-resolution signals. Unlike assimilation-dependent approaches, our method avoids $XCO_2$ reanalysis inputs, preserving satellite-scale fidelity through high-resolution environmental variables modeling while maintaining precision."

[Figure]

**Figure 13.** Comparison between the OCO-2 $XCO_2$ data, accessible $XCO_2$ products from Wang et al. (2023), Sheng et al. (2022), Zhang et al. (2023), and our reconstructed $XCO_2$ data in four regions, using the products of January of 2015 as an example.

**Q3.** The study lacks specificity in demonstrating how the new dataset quantitatively improves understanding relative to existing satellite data. Providing explicit examples or quantifiable differences would enhance the significance of this study.

**Response:** Thanks for this suggestion. Compared with the existing satellite data and reconstructed data, our products deliver two major enhancements: (1) Our reconstructed $XCO_2$ product overcomes the extensive data gaps typically caused by narrow satellite swaths and retrieval interference from clouds and aerosols, achieving complete global coverage without compromising measurement accuracy. (2) Relative to existing publicly available full-coverage global $XCO_2$ products, our product offers the finest spatial resolution (0.05°). Moreover, our method avoids coarse $XCO_2$ reanalysis inputs, preserving satellite-scale fidelity through high-resolution environmental variables modeling. Consequently, the products enable enhanced spatial details in identifying regional- and county-level $XCO_2$ hotpots, carbon emissions and fragmented carbon sinks, providing a robust basis for targeted global carbon governance policies at relevant scales. We have added explicit comparison between

OCO-2/3 data and our products in section **3.2 Spatiotemporal pattern of global XCO₂** as follow:

"The global distribution of annual mean $XCO_2$ concentration from 2015 to 2021 is illustrated in Fig. 8. The results reveal pronounced spatial heterogeneity in $XCO_2$ concentrations, characterized by a marked hemispheric asymmetry. Specifically, the Northern Hemisphere exhibited systematically elevated $XCO_2$ levels compared to the Southern Hemisphere, consistent with latitudinal gradients driven by anthropogenic emission patterns and atmospheric transport dynamics. Regionally, North America, East Asia, Central Africa, and northwest of Southern America were identified as persistent hotspots of enhanced $XCO_2$. The high concentrations of $XCO_2$ in North America and East Asia stem primarily from the fossil fuel emission from energy production and transportation sectors. Whereas the tropical regions (i.e., Central Africa and South America) are influenced by coupled biomass burning and land-use changes.

[Figure]

**Figure 8.** The global spatial distribution of reconstructed annual mean $XCO_2$ concentration from 2015 to 2021.

We also provided the annual OCO-2 $XCO_2$ data from 2015 to 2019 and OCO-3 $XCO_2$ data from 2020 to 2021 in Fig. 9. Spatially, our reconstructed $XCO_2$ dataset (Fig. 8) demonstrates robust consistency with satellite observations, particularly in midlatitude industrialized regions where both datasets capture emission hotspots. Notably, OCO-3 exhibits denser observational sampling due to its improved spatial coverage and swath width compared to OCO-2's narrow tracks. However, persistent data gaps remain prevalent in both two satellite products after annual aggregating. These spatial coverage limitations hinder fine-scale global analysis, particularly in assessing localized emission sources and regional scale carbon flux."

[Figure]

**Figure 9.** The global spatial distribution of annual mean OCO-2/OCO-3 $XCO_2$ concentration from 2015 to 2021.

Additionally, we also added a detailed local-scale evaluation contrasting OCO-2/3 observations, our reconstructed $XCO_2$ product, and other publicly available global $XCO_2$ datasets in **4.1 Comparison with previous studies**, as follows:

"To evaluate the advancement of our $XCO_2$ product, we compared it with original OCO-2 observations and publicly available global $XCO_2$ datasets (Wang et al., 2023; Sheng et al., 2022; Zhang et al., 2023) across four regions: North America, Europe with northern Africa, Asia, and Oceania (Fig. 13) in January 2015. Despite monthly aggregation, OCO-2 data exhibit persistent spatial discontinuities, limiting the capacity to analyze monthly $XCO_2$ variability at regional and national scales. Existing $XCO_2$ products (spatial resolution of 0.25°, 1°, and 0.1°, respectively) broadly reproduce large-scale $XCO_2$ patterns but fail to resolve fine-scale heterogeneity. In comparison,

our reconstructed XCO₂, with the highest spatial resolution, provides a more detailed and accurate representation of the regional $XCO_2$ patterns. For example, lower $XCO_2$ concentrations are clearly identified in eastern Canada (The first row of Fig.13) and Papua New Guinea (The fourth row of Fig. 13), regions characterized by dense forest cover. This correspondence highlights the substantial carbon sink potential of these forested areas. Our high-resolution product better identifies the $CO_2$ heterogeneity associated with different land cover types, whereas the coarse-resolution products smooth these signals. This limitation primarily stems from the neglect of high-resolution land cover dynamics and dependence on coarse-resolution assimilated/reanalysis datasets (e.g., CAMS $XCO_2$, CarbonTracker), resulting in oversmoothed spatial patterns that obscure satellite-derived high-resolution signals. Unlike assimilation-dependent approaches, our method avoids $XCO_2$ reanalysis inputs, preserving satellite-scale fidelity through high-resolution environmental variables modeling while maintaining precision."

[Figure]

**Figure 13.** Comparison between the OCO-2 $XCO_2$ data, accessible $XCO_2$ products from Wang et al. (2023), Sheng et al. (2022), Zhang et al. (2023), and our reconstructed $XCO_2$ data in four regions, using the products of January of 2015 as an example.

**Q4.** The conclusion stating "promising advancement" is too broad. It should be explicitly clarified what specific policy, modeling, or scientific implications this advancement has, thus highlighting concrete applications or benefits.

**Response:** Thanks for this constructive suggestion. We have revised the section **6. Conclusion** and explicitly clarified the key contributions of this study:

"The main conclusions and contributions are as follows:

(1) The advanced At-BiLSTM model could successfully established the nonlinear relationship between satellite-derived $XCO_2$ and a set of key environmental variables. And the reconstructed $XCO_2$ based on our model shows relatively good agreement with TCCON $XCO_2$, with $R^2$, RMSE, and MAE values of 0.91, 1.58 ppm, and 1.22 ppm, respectively.

(2) The reconstructed $XCO_2$ product overcomes the extensive data gaps typically caused by narrow satellite swaths and retrieval interference from clouds and aerosols, achieving complete global coverage. Moreover, relative to existing publicly available full-coverage $XCO_2$ datasets, our product offers the finest spatial resolution (0.05°) while maintaining comparable accuracy.

(3) Our method avoids coarse $XCO_2$ reanalysis inputs, preserving satellite-scale fidelity through high-resolution environmental variables modeling. Consequently, the products enable enhanced ability in identifying regional- and county-level $XCO_2$ hotpots, carbon emissions and fragmented carbon sinks, providing a robust basis for targeted global carbon governance policies."

We also changed the description in the **Abstract**:

"The $XCO_2$ dataset is publicly accessible on the Zenodo platform at https://doi.org/10.5281/zenodo.12706142 (Wang et al., 2024). Our products enable enhanced ability in identifying regional- and county-level $XCO_2$ hotpots, carbon emissions and fragmented carbon sinks, providing a robust basis for targeted global carbon governance policies."

---

## Author Response (AR1)

**Cover Letter**

Dear Editors and Referees,

On behalf of all co-authors, I thank you for your and for the anonymous referees'

comments on our manuscript entitled "A full-coverage satellite-based global

atmospheric CO2 dataset at 0.05° resolution from 2015 to 2021 for exploring global

carbon dynamics". We appreciate the positive and constructive comments. We

carefully considered the comments and suggestions point-by-point and modified the

manuscript accordingly.

Please note that we used blue text for our responses to the comments and red text to

show changes in the revised manuscript. Overall, this manuscript has been re-edited

and proofread by all the authors.

We hope the revised version of the manuscript will be considered for publication in

Earth System Science Data.

Should you have any questions, please do not hesitate to contact me or any of the coauthors. I am looking forward to hearing from you.

Thank you very much.

Sincerely,

Dr. Zhige Wang (on behalf of all co-authors)

College of Environmental and Resource Sciences

Zhejiang University,

Hangzhou 310058, China

E-mail: zgwang@zju.edu.cn

**Response to Anonymous Referee #1:**

This manuscript presents the development of a global dataset of column-averaged dry-air mole fraction of CO2 (XCO2) at high resolution (0.05°) using multi satellite products, and an improved deep learning model. They further evaluate new datasets using the measurements from the TCCON network. While the study shows promise, there are significant concerns regarding methodological transparency, and clarity of explicit demonstration of advantages over existing satellite products. Addressing these issues will greatly enhance the manuscript's impact and originality.

**Response:** We sincerely appreciate your thorough review of our manuscript and the valuable, constructive feedback. In response, we have expanded our discussion of the methods, and the advantages of our products over existing XCO2 data. Our point-by-point responses to your comments are provided below.

**Q1.** (1) The manuscript lacks in describing the methodological details of the improved deep learning model. The authors should clearly outline the specific innovations or modifications that lead to improved accuracy.

**Response:** Thanks for the suggestion. Given the complex temporal dependencies and nonlinear relationships between atmospheric XCO2 and a wide range of environmental variables, we selected the Attention-based Bidirectional Long Short-Term Memory (At-BiLSTM) model for this study. This choice is motivated by several key considerations:

Firstly, LSTM networks are well-suited for modeling temporal sequences and capturing long-range dependencies, which is essential for understanding the seasonal variations of XCO2 and dynamic feedbacks between XCO2 and environmental drivers such as temperature, vegetation activity, and surface pressure. The bidirectional structure enhances this capability by allowing the model to consider both past and future context in the time series, thereby providing a more comprehensive representation of the underlying temporal dynamics.

Secondly, the incorporation of the attention mechanism enables the model to dynamically focus on the most critical time steps when making predictions. This is particularly important when dealing with high-dimensional input data comprising multi-timestep variables, as it allows the model to assign different weights to different input features, thereby improving interpretability and predictive performance.

Finally, the At-BiLSTM model's ability to capture nonlinear relationships is crucial in the context of atmospheric CO2 modeling, where interactions between variables are complex and nonlinear. By leveraging the strengths of deep learning, the model can learn intricate patterns from the multi-source data that are difficult to capture with traditional statistical or linear models.

Therefore, we chose At-BiLSTM model as a robust and flexible framework to reconstructing XCO2 at fine spatial resolution with improved accuracy and spatiotemporal consistency.

We have included the necessary clarifications in 2.2 Deep learning-based XCO2 reconstruction:

"The LSTM model is a variant of RNN that excels in modeling temporal sequences and capture long-range dependencies (Hochreiter and Schmidhuber, 1997; Graves et al., 2005), which is essential for understanding the seasonal variations of XCO2 and dynamic feedbacks between XCO2 and environmental drivers we selected. Each LSTM cell includes an input gate, a forget gate and an output gate. The forget gate  $f_t$  determines which information from the previous time step to forget (Eq. 1):

$$f_t = \sigma(W_f \cdot [h_{t-1}, x_t] + b_f) \tag{1}$$

where  $\sigma$ ,  $W_f$ ,  $[h_{t-1}, x_t]$ , and  $b_f$  denotes the sigmoid activation function, vectors of weights, concatenation of the hidden state at timestep t-1 and the current input, and the bias vector, respectively.

The input gate  $i_t$  governs the selective storage of the data in current time step, and the output from forget gate  $f_t$  and input gate  $i_t$  are combined in the cell state  $C_t$  (Eq. 2-3):

$$i_t = \sigma(W_i \cdot [h_{t-1}, x_t] + b_i) \tag{2}$$

$$C_t = f_t \cdot C_{t-1} + i_t \cdot \tanh(W_C \cdot [h_{t-1}, x_t] + b_C)$$
(3)

where  $W_i$  and  $W_C$  denote the weight matrix for the input gate and the current cell state, respectively;  $b_i$  and  $b_c$  are the bias vector of the input gate and the current cell state, respectively;  $C_{t-1}$  and tanh represent the cell state at timestep t-1 and the activation function.

Lastly, the output gate  $o_t$  controls the flow of information from the cell state to the next time step.

$$o_t = \sigma(W_o \cdot [h_{t-1}, x_t] + b_o) \tag{4}$$

where  $W_o$  and  $b_o$  denotes the weight matrix and the bias vector of the output gate, respectively.

These gate structures effectively manage the flow of information within the LSTM, enabling it to capture the temporal dependencies present in the data (Yuan et al., 2020; Su et al., 2021). Bidirectional LSTM consists of two directional LSTM, in which the data flows forward and backward (Graves et al., 2013). The bidirectional structure was chosen to enhance the capability of LSTM by allowing the model to consider both past and future context in the time series, thereby providing a more comprehensive understanding of the underlying temporal dynamics.

We also defined a multi-dimensional attention layer behind the BiLSTM to focus on more critical timesteps and give them higher weights (Bahdanau et al., 2016). This is particularly important when dealing with high-dimensional input data comprising multi-timestep variables, as it allows the model to assign different weights to different timesteps, thereby improving interpretability and predictive performance (Liu and Guo, 2019; Wang et al., 2024b). Based on this framework, the At-BiLSTM model offers a robust and flexible framework for linking XCO2 with multiple environmental variables and reconstructing XCO2 at fine spatial resolution with improved accuracy and spatiotemporal consistency." (Page 11-12 Line 262-298)

And we have also added the detailed deployment and output of this deep learning model as follows:

"The At-BiLSTM consists of one input layer, three Bidirectional LSTM (Bi-LSTM) layers, one attention layer, one dropout layer to prevent overfitting, and one fully connected layer (i.e., dense layer) for the final output. Each Bi-LSTM includes 512 hidden units and tanh activation in both forward and backward directions. The attention mechanism learns a weight distribution over the time dimension using a Dense layer with softmax activation, then multiplies these weights with the BiLSTM output to emphasize important time steps. The detailed deployment and output are provided in Table 3. The model was implemented using the TensorFlow and Keras deep learning APIs in Python. A time step of 3 was used, and the model was trained for 200 epochs with the mean squared error (MSE) as the loss function. A step-wise decay strategy was applied to the learning rate, where the rate was reduced by a factor of 10 every 50 epochs to enhance training stability and convergence. Prior to training, all input data were normalized using the mean and standard deviation of the dataset."

Table 3. Architecture of the At-BiLSTM model

| Layer Name   | er Name Layer Parameters |                                   |          |
|--------------|--------------------------|-----------------------------------|----------|
| Bi-LSTM      | Input layer              |                                   | 3×16     |
|              | Bi-LSTM1                 | units = 512, activation = 'tanh'  | 3×1024   |
|              | Bi-LSTM2                 | units = 512, activation = 'tanh'  | 3 × 1024 |
|              | Bi-LSTM3                 | units = 512, activation = 'tanh'  | 3×1024   |
| Attention    | Permute                  | -                                 | 1024×3   |
|              | Dense                    | units = 3, activation = 'softmax' | 1024×3   |
|              | Permute                  | -                                 | 3×1024   |
|              | Multiply                 | -                                 | 3×1024   |
| Dropout      |                          | rate = 0.5                        |          |
| Full-connect | Dense                    | units = 1                         | 1        |
| 10 10 T      | 200 212)                 |                                   |          |

(Page 12-13 Line 299-313)

(2) Additionally, since the formation and distribution of XCO2 are influenced by atmospheric transport processes across multiple vertical layers (and not solely by surface fluxes), it is important that the manuscript explains how these vertical transport processes are incorporated into the model. If these processes are not accounted for, this limitation should be explicitly acknowledged.

Response: Thank you for raising this important point. In this study, we estimated column-averaged CO2 (XCO2), and as such, vertical transport processes were not explicitly incorporated into the modeling framework. However, we acknowledge that vertical redistribution of CO2 through atmospheric transport processes (e.g., mixing and convection) can significantly influence the spatiotemporal patterns of XCO2, particularly by altering the linkage between surface fluxes and column concentrations. The absence of vertical transport indicators in our model may limit its accuracy, especially in regions or periods characterized by strong vertical mixing. We have included a discussion of this limitation and have highlighted it as a key area for future model enhancement in 4.2 Limitations and future improvements:

"Additionally, though our model integrates multiple environmental variables associated with surface carbon flux variations, it does not account for vertical atmospheric transport. As XCO2 represents the column-averaged CO2 concentration, vertical redistribution of CO2 through atmospheric transport (e.g., mixing, convection)

can alter the relationship between surface carbon fluxes and column concentrations. The absence of such vertical transport indicators may reduce the model's accuracy in regions or periods with strong vertical mixing. Future efforts will incorporate vertical transport-related variables, such as planetary boundary layer height, vertical wind components, and other reanalysis-derived indicators, to better represent the atmospheric processes that influence the column-averaged CO2 signal." (Page 25-26 Line 553-562)

Q2. Although the authors present a new global XCO2 product at 0.05° resolution, the distinct novelty and advantages over existing datasets remain unclear. The study should explicitly state how the analyzed information significantly differs from or improves upon existing satellite data. While validation against TCCON is good, the authors should explicitly compare these results with those from existing datasets to clearly demonstrate accuracy improvements.

**Response:** Thanks for the constructive suggestion. To validate the effectiveness of our model and resulting XCO2 products, we firstly compared our results with current studies which focuses on global XCO2 reconstruction. Afterwards, we compared our products with original OCO-2 observations and three publicly available global XCO2 datasets to evaluate the advancement of our XCO2 product. The comparison results have added in **4.1 Comparison with previous studies** as follow:

"To validate the effectiveness of our model and resulting XCO2 products, we compared our results with current studies which focuses on global XCO2 reconstruction (Table 5). As for the in-situ validation, most existing studies report high accuracy with almost all R2 over 0.9, RMSE less than 2 ppm. Regarding spatial resolution, the various products differ substantially, ranging from 1° down to 0.01°. It should be noted that increasing spatial resolution tends to compromise the accuracy of XCO2 retrievals. However, our XCO2 product achieves an optimal balance between spatial detail and measurement precision, exhibiting both high spatial resolution (0.05°) and robust accuracy (R2=0.91, RMSE =1.54 ppm) in comprehensive evaluations.

**Table 5.** Comparison between current studies focusing on global XCO2 reconstruction

| Model                  | Spatial resolution | In-situ validation (with TCCON) |            | Reference |                     |
|------------------------|--------------------|---------------------------------|------------|-----------|---------------------|
|                        |                    | R 2                  | RMSE (ppm) | MAE (ppm) | -                   |
| Attentional-based LSTM | 0.05°              | 0.91                            | 1.54       | 1.22      | Our study           |
| Deep forest            | 0.1°               | 0.96                            | 1.01       | -         | Zhang et al. (2023) |
| S-STDCT                | 0.25°              | 0.95                            | 1.18       | -         | Wang et al. (2023)  |
| Spatiotemporal kriging | 1°                 | 0.97                            | 1.13       | 0.88      | Sheng et al. (2022) |
| MLE & OI               | $0.5^{\circ}$      | 0.92                            | 2.62       | 1.53      | Jin et al. (2022)   |
| ERT                    | 0.01°              | 0.83                            | 1.79       | -         | Li et al. (2022)    |

\*S-STDCT: Self-supervised spatiotemporal discrete cosine transform, MLE & OI: maximum likelihood estimation method and optimal interpolation; ERT: Extremely randomized trees

To evaluate the advancement of our XCO2 product, we compared it with original OCO-2 observations and publicly available global XCO2 datasets (Wang et al., 2023; Sheng et al., 2022; Zhang et al., 2023) across four regions: North America, Europe with northern Africa, Asia, and Oceania (Fig. 14) in January 2015. Despite monthly aggregation, OCO-2 data exhibit persistent spatial discontinuities, limiting the capacity to analyze monthly XCO2 variability at regional and national scales. Existing XCO2 products (spatial resolution of 0.25°, 1°, and 0.1°, respectively) broadly reproduce large-scale XCO2 patterns but fail to resolve fine-scale heterogeneity. In comparison, our reconstructed XCO2, with the highest spatial resolution, provides a more detailed and accurate representation of the regional XCO2 patterns. For example, lower XCO2 concentrations are clearly identified in eastern Canada (The first row of Fig.14) and Papua New Guinea (The fourth row of Fig. 14), regions characterized by dense forest cover. This correspondence highlights the substantial carbon sink potential of these forested areas. Our high-resolution product better identifies the CO2 heterogeneity associated with different land cover types, whereas the coarse-resolution products smooth these signals. This limitation primarily stems from the neglect of highresolution land cover dynamics and dependence on coarse-resolution assimilated/reanalysis datasets (e.g., CAMS XCO2, CarbonTracker), resulting in oversmoothed spatial patterns that obscure satellite-derived high-resolution signals. Unlike assimilation-dependent approaches, our method avoids XCO2 reanalysis inputs, preserving satellite-scale fidelity through high-resolution environmental variables modeling while maintaining precision."

**Figure 14.** Comparison between the OCO-2 XCO2 data, accessible XCO2 products from Wang et al. (2023), Sheng et al. (2022), Zhang et al. (2023), and our reconstructed XCO2 data in four regions, using the products of January of 2015 as an example.

**(Page 23-25 Line 504-542)**

**Q3.** The study lacks specificity in demonstrating how the new dataset quantitatively improves understanding relative to existing satellite data. Providing explicit examples or quantifiable differences would enhance the significance of this study.

Response: Thanks for this suggestion. Compared with the existing satellite data and reconstructed data, our products deliver two major enhancements: (1) Our reconstructed XCO2 product overcomes the extensive data gaps typically caused by narrow satellite swaths and retrieval interference from clouds and aerosols, achieving complete global coverage without compromising measurement accuracy. (2) Relative to existing publicly available full-coverage global XCO2 products, our product offers the finest spatial resolution (0.05°). Moreover, our method avoids coarse XCO2 reanalysis inputs, preserving satellite-scale fidelity through high-resolution environmental variables modeling. Consequently, the products enable enhanced spatial details in identifying regional- and county-level XCO2 hotpots, carbon emissions and fragmented carbon sinks, providing a robust basis for targeted global carbon governance policies at relevant scales. We have added explicit comparison between

**OCO-2/3 data and our products in section **3.2 Spatiotemporal pattern of global XCO2** as follow:**

"The global distribution of annual mean XCO2 concentration from 2015 to 2021 is illustrated in Fig. 9. The results reveal pronounced spatial heterogeneity in XCO2 concentrations, characterized by a marked hemispheric asymmetry. Specifically, the Northern Hemisphere exhibited systematically elevated XCO2 levels compared to the Southern Hemisphere, consistent with latitudinal gradients driven by anthropogenic emission patterns and atmospheric transport dynamics. Regionally, North America, East Asia, Central Africa, and northwest of Southern America were identified as persistent hotspots of enhanced XCO2. The high concentrations of XCO2 in North America and East Asia stem primarily from the fossil fuel emission from energy production and transportation sectors. Whereas the tropical regions (i.e., Central Africa and South America) are influenced by coupled biomass burning and land-use changes.

**Figure 9.** The global spatial distribution of reconstructed annual mean XCO2 concentration from 2015 to 2021.

We also provided the annual OCO-2 XCO2 data from 2015 to 2019 and OCO-3 XCO2 data from 2020 to 2021 in Fig. 10. Spatially, our reconstructed XCO2 dataset (Fig. 9) demonstrates robust consistency with satellite observations, particularly in midlatitude industrialized regions where both datasets capture emission hotspots. Notably, OCO-3 exhibits denser observational sampling due to its improved spatial coverage and swath width compared to OCO-2's narrow tracks. However, persistent data gaps remain prevalent in both two satellite products after annual aggregating. These spatial coverage limitations hinder fine-scale global analysis, particularly in assessing localized emission sources and regional scale carbon flux."

**Figure 10.** The global spatial distribution of annual mean OCO-2/OCO-3 XCO2 concentration from 2015 to 2021.

**(Page 18-19 Line 404-429)**

Additionally, we also added a detailed local-scale evaluation contrasting OCO-2/3 observations, our reconstructed XCO2 product, and other publicly available global XCO2 datasets in **4.1 Comparison with previous studies**, as follows:

"To evaluate the advancement of our XCO2 product, we compared it with original OCO-2 observations and publicly available global XCO2 datasets (Wang et al., 2023; Sheng et al., 2022; Zhang et al., 2023) across four regions: North America, Europe with northern Africa, Asia, and Oceania (Fig. 14) in January 2015. Despite monthly aggregation, OCO-2 data exhibit persistent spatial discontinuities, limiting the capacity to analyze monthly XCO2 variability at regional and national scales. Existing XCO2 products (spatial resolution of 0.25°, 1°, and 0.1°, respectively) broadly reproduce

large-scale XCO2 patterns but fail to resolve fine-scale heterogeneity. In comparison, our reconstructed XCO2, with the highest spatial resolution, provides a more detailed and accurate representation of the regional XCO2 patterns. For example, lower XCO2 concentrations are clearly identified in eastern Canada (The first row of Fig.14) and Papua New Guinea (The fourth row of Fig. 14), regions characterized by dense forest cover. This correspondence highlights the substantial carbon sink potential of these forested areas. Our high-resolution product better identifies the CO2 heterogeneity associated with different land cover types, whereas the coarse-resolution products smooth these signals. This limitation primarily stems from the neglect of highresolution land cover dynamics and dependence on coarse-resolution assimilated/reanalysis datasets (e.g., CAMS XCO2, CarbonTracker), resulting in oversmoothed spatial patterns that obscure satellite-derived high-resolution signals. Unlike assimilation-dependent approaches, our method avoids XCO2 reanalysis inputs, preserving satellite-scale fidelity through high-resolution environmental variables modeling while maintaining precision."

**Figure 14.** Comparison between the OCO-2 XCO2 data, accessible XCO2 products from Wang et al. (2023), Sheng et al. (2022), Zhang et al. (2023), and our reconstructed XCO2 data in four regions, using the products of January of 2015 as an example.

(Page 23-25 Line 517-542)

**Q4.** The conclusion stating "promising advancement" is too broad. It should be explicitly clarified what specific policy, modeling, or scientific implications this advancement has, thus highlighting concrete applications or benefits.

**Response:** Thanks for this constructive suggestion. We have revised the section **6.**Conclusion and explicitly clarified the key contributions of this study:

"The main conclusions and contributions are as follows:

- (1) The advanced At-BiLSTM model could successfully established the nonlinear relationship between satellite-derived XCO2 and a set of key environmental variables. And the reconstructed XCO2 based on our model shows relatively good agreement with TCCON XCO2, with R2, RMSE, and MAE values of 0.91, 1.58 ppm, and 1.22 ppm, respectively.
- (2) The reconstructed XCO2 product overcomes the extensive data gaps typically caused by narrow satellite swaths and retrieval interference from clouds and aerosols, achieving complete global coverage. Moreover, relative to existing publicly available full-coverage XCO2 datasets, our product offers the finest spatial resolution (0.05°) while maintaining comparable accuracy.
- (3) Our method avoids coarse XCO2 reanalysis inputs, preserving satellite-scale fidelity through high-resolution environmental variables modeling. Consequently, the products enable enhanced ability in identifying regional- and county-level XCO2 hotpots, carbon emissions and fragmented carbon sinks, providing a robust basis for targeted global carbon governance policies." (Page 27 Line 594-610)

We also changed the description in the **Abstract**:

"The XCO2 dataset is publicly accessible on the Zenodo platform at <a href="https://doi.org/10.5281/zenodo.12706142">https://doi.org/10.5281/zenodo.12706142</a> (Wang et al., 2024a). Our products enable enhanced ability in identifying regional- and county-level XCO2 hotpots, carbon emissions and fragmented carbon sinks, providing a robust basis for targeted global carbon governance policies." (Page 2 Line 46-51)

**Response to Anonymous Referee #2:**

This study reconstructed a global full-coverage XCO2 product with a 0.05° spatial resolution using multi-component satellite data and an advanced deep learning method. However, the manuscript lacks innovation and sufficient detail in several aspects. My comments are listed below:

**Response:** We are truly grateful for reviewing our manuscript and providing us with constructive feedback. Considering your feedback, we have elaborated on the innovation of our study and provided additional details in section **2. materials and methods**, **3. Results** and **4. Discussion**. We responded to your comments point by point as below.

**Q1.** (1) The manuscript notes that the spatial resolution of current global full-coverage XCO2 products is relatively coarse, ranging from approximately 0.25° to 2° (Line 128). However, global XCO2 products with a 0.1° spatial resolution already exist (https://doi.org/10.1016/j.envint.2023.108057), indicating a need for more comprehensive literature review.

**Response:** Thank you for pointing out this oversight and we apologize for omitting this important reference. We have provided a more tailed literature review in 1. introduction as follows:

"The second category is regression-based methods, which aim to fill the gap by capturing the nonlinear relationship between multi-source XCO2 measurements and related covariates (He et al., 2022; Siabi et al., 2019; Zhang and Liu, 2023). The specific methods include traditional statistical models, geostatistical models and machine learning models. Siabi et al. (2019) employed the Artificial Neural Network (ANN) to establish correlation between XCO2 and eight environmental variables. Zhang and Liu (2023) utilized the convolution neural networks (CNN) coupled with attention mechanisms to produce full-coverage XCO2 data across China. Recently, Zhang et al. (2023) developed high spatial resolution global CO2 concentration data based on deep forest model and multi-source satellite products.

Although the development of CO2 observation satellites and the application of machine learning methods have significantly improved the estimation accuracy of

XCO2, current studies still face several limitations. Firstly, due to the sparse distribution of satellite XCO2 data, previous studies always relied on assimilation and reanalysis XCO2 data, such as CAMS XCO2 with coarse spatial resolution (0.75°). This reliance often results in final products that closely mirror the assimilation and reanalysis results, leading to an oversmoothed distribution that undermines the high-resolution advantages of satellite data. Furthermore, most current studies estimated the spatial distribution of CO2 primarily based on vegetation and meteorological information, with limited consideration of the impact of human activities and emissions, despite these have significant influence on atmospheric CO2 variability. This limitation also led to estimation results that fail to adequately capture the impact of anthropogenic emissions on atmospheric CO2. In addition, most studies that employ regression models to estimate full-coverage XCO2 are limited to regional or national scales due to the weak transferability of these models. Only a few studies (Zheng et al., 2023) have explored global-scale CO2 estimation using machine learning approaches, highlighting the need for further research to enhance model generalizability and scalability. Therefore, we intent to develop the global full-coverage XCO2 products with the capacity to capture both large-scale patterns and fine spatial details. This development leveraged satellite carbon monitoring, multi-source high spatial resolution auxiliary variables and advanced methods that exhibit spatiotemporal transferability to overcome the aforementioned limitations." (Page 4-5 Line 106-138)

(2) Although this study improves the XCO2 spatial resolution to 0.05°, its innovation and advantages compared to other datasets remain unclear. It is recommended to clearly articulate the study's novelty and specific strengths.

**Response:** Thanks for this constructive suggestion. We added a more detailed comparison with other datasets to highlight the innovation and advantages of our study in section **4.1 Comparison with previous studies**, as follows:

"To validate the effectiveness of our model and resulting XCO2 products, we compared our results with current studies which focuses on global XCO2 reconstruction (Table 5). As for the in-situ validation, most existing studies report high accuracy with almost all R2 over 0.9, RMSE less than 2 ppm. Regarding spatial resolution, the various products differ substantially, ranging from 1° down to 0.01°. It should be noted that

increasing spatial resolution tends to compromise the accuracy of  $XCO_2$  retrievals. However, our  $XCO_2$  product achieves an optimal balance between spatial detail and measurement precision, exhibiting both high spatial resolution  $(0.05^{\circ})$  and robust accuracy ( $R^2$ =0.91, RMSE =1.54 ppm) in comprehensive evaluations.

Table 5. Comparison between current studies focusing on global XCO2 reconstruction

| Model                  | Spatial resolution | In-situ validation (with TCCON) |            |           | Reference           |
|------------------------|--------------------|---------------------------------|------------|-----------|---------------------|
|                        |                    | R 2                  | RMSE (ppm) | MAE (ppm) |                     |
| Attentional-based LSTM | 0.05°              | 0.91                            | 1.54       | 1.22      | Our study           |
| Deep forest            | 0.1°               | 0.96                            | 1.01       | -         | Zhang et al. (2023) |
| S-STDCT                | 0.25°              | 0.95                            | 1.18       | -         | Wang et al. (2023)  |
| Spatiotemporal kriging | 1°                 | 0.97                            | 1.13       | 0.88      | Sheng et al. (2022) |
| MLE & OI               | 0.5°               | 0.92                            | 2.62       | 1.53      | Jin et al. (2022)   |
| ERT                    | 0.01°              | 0.83                            | 1.79       | -         | Li et al. (2022)    |

\*S-STDCT: Self-supervised spatiotemporal discrete cosine transform; MLE & OI: maximum likelihood estimation method and optimal interpolation; ERT: Extremely randomized trees

To evaluate the advancement of our XCO2 product, we compared it with original OCO-2 observations and publicly available global XCO2 datasets (Wang et al., 2023; Sheng et al., 2022; Zhang et al., 2023) across four regions: North America, Europe with northern Africa, Asia, and Oceania (Fig. 14) in January 2015. Despite monthly aggregation, OCO-2 data exhibit persistent spatial discontinuities, limiting the capacity to analyze monthly XCO2 variability at regional and national scales. Existing XCO2 products (spatial resolution of 0.25°, 1°, and 0.1°, respectively) broadly reproduce large-scale XCO2 patterns but fail to resolve fine-scale heterogeneity. In comparison, our reconstructed XCO2, with the highest spatial resolution, provides a more detailed and accurate representation of the regional XCO2 patterns. For example, lower XCO2 concentrations are clearly identified in eastern Canada (The first row of Fig.14) and Papua New Guinea (The fourth row of Fig. 14), regions characterized by dense forest cover. This correspondence highlights the substantial carbon sink potential of these forested areas. Our high-resolution product better identifies the CO2 heterogeneity associated with different land cover types, whereas the coarse-resolution products smooth these signals. This limitation primarily stems from the neglect of highland cover dynamics and dependence on coarse-resolution assimilated/reanalysis datasets (e.g., CAMS XCO2, CarbonTracker), resulting in oversmoothed spatial patterns that obscure satellite-derived high-resolution signals. Unlike assimilation-dependent approaches, our method avoids XCO2 reanalysis inputs, preserving satellite-scale fidelity through high-resolution environmental variables modeling while maintaining precision.

**Figure 14.** Comparison between the OCO-2 XCO2 data, accessible XCO2 products from Wang et al. (2023), Sheng et al. (2022), Zhang et al. (2023), and our reconstructed XCO2 data in four regions, using the products of January of 2015 as an example." (Page 23-25 Line 504-542)

We also summarized the advantages and contributions of our study in **6. Conclusion** as follows:

"As a major driver of global warming, the monitoring of CO2 changes, especially anthropogenic CO2 emissions, is of critical importance. The launch of carbon satellites offers a significant advancement for CO2 monitoring. However, the limited spatial coverage of satellite observations constrains the utility of XCO2 data. While current XCO2 products exhibit relatively high validation accuracy, their coarse spatial resolution remains inadequate for applications such as regional- or county-level emission monitoring, as well as for the detection and inversion of large emission sources. To address these issues, we reconstructed a global full-coverage XCO2 product at a fine spatial resolution of 0.05° and temporal resolution of 1 month from 2015 to 2021. The advanced deep learning method was adopted to model time-series XCO2 and

incorporate terrestrial flux, anthropogenic flux and climatic impacts into the parameterization process. Through comprehensive evaluations, including cross-validation, in-situ validation, spatial distribution assessment and comparison with other XCO2 products, our reconstructed XCO2 products demonstrates significant improvements in both accuracy and spatial resolution. The main conclusions and contributions are as following:

- (1) The advanced At-BiLSTM model could successfully established the nonlinear relationship between satellite-derived XCO2 and a set of key environmental variables. And the reconstructed XCO2 based on our model shows relatively good agreement with TCCON XCO2, with R2, RMSE, and MAE values of 0.91, 1.58 ppm, and 1.22 ppm, respectively.
- (2) The reconstructed XCO2 product overcomes the extensive data gaps typically caused by narrow satellite swaths and retrieval interference from clouds and aerosols, achieving complete global coverage. Moreover, relative to existing publicly available full-coverage XCO2 datasets, our product offers the finest spatial resolution (0.05°) while maintaining comparable accuracy.
- (3) Our method avoids coarse XCO2 reanalysis inputs, preserving satellite-scale fidelity through high-resolution environmental variables modeling. Consequently, the products enable enhanced ability in identifying regional- and county-level XCO2 hotpots, carbon emissions and fragmented carbon sinks, providing a robust basis for targeted global carbon governance policies." (Page 26-27 Line 580-610)

And we have further elaborated on the specific strength of our products in 1. **Introduction** as follows:

"In this study, we leveraged time-series OCO-2/3 XCO2 data and various related environmental variables from multi-source satellites to generate global full-coverage XCO2 products. The advanced deep learning method was adopted to model time-series XCO2 and incorporate terrestrial flux, anthropogenic flux and climatic impacts into the parameterization process. These products are designed to meet the following criteria: (1) high validated accuracy to ensure the reliability of the estimates, (2) high spatial resolution capable of capturing fine-scale variations in CO2 concentrations, and (3) global full-coverage that overcomes missing values in satellite carbon observations." (Page 5 Line 139-146)

**Q2.** The model methodology section lacks essential explanations. The study employs the Attention-based Bidirectional Long Short-Term Memory (At-BiLSTM) model for global XCO2 reconstruction, but it does not justify the choice of this model or clarify its advantages over traditional LSTM models. Additionally, the model's interpretation remains unclear. It is recommended to provide a rationale for selecting At-BiLSTM and elucidate its specific benefits and interpretive framework.

**Response:** Thanks for this constructive suggestion. Given the complex temporal dependencies and nonlinear relationships between atmospheric XCO2 and a wide range of environmental variables, we selected the Attention-based Bidirectional Long Short-Term Memory (At-BiLSTM) model for this study. This choice is motivated by several key considerations:

Firstly, LSTM networks are well-suited for modeling temporal sequences and capturing long-range dependencies, which is essential for understanding the seasonal variations of XCO2 and dynamic feedbacks between XCO2 and environmental drivers such as temperature, vegetation activity, and surface pressure. The bidirectional structure enhances this capability by allowing the model to consider both past and future context in the time series, thereby providing a more comprehensive representation of the underlying temporal dynamics.

Secondly, the incorporation of the attention mechanism enables the model to dynamically focus on the most critical time steps when making predictions. This is particularly important when dealing with high-dimensional input data comprising multi-timestep variables, as it allows the model to assign different weights to different input features, thereby improving interpretability and predictive performance.

Finally, the At-BiLSTM model's ability to capture nonlinear relationships is crucial in the context of atmospheric CO2 modeling, where interactions between variables are complex and nonlinear. By leveraging the strengths of deep learning, the model can learn intricate patterns from the multi-source data that are difficult to capture with traditional statistical or linear models.

Therefore, we chose At-BiLSTM model as a robust and flexible framework to reconstructing XCO2 at fine spatial resolution with improved accuracy and spatiotemporal consistency.

We have included the necessary clarifications of its advancement in 2.2 Deep learning-based XCO2 reconstruction:

[revised manuscript text omitted]

**(Page 12-13 Line 299-313)**

**Q3.** (1) The discussion section requires further elaboration. It should comprehensively address the advantages of the model used and the resulting full-coverage XCO2 product compared to other models and datasets.

**Response:** Thank you for this valuable comment. We have revised the discussion section, and added two sub-section: **4.1 Comparison with previous studies** and **4.2 Limitations and future improvements**. In added section **4.1 Comparison with previous studies**, we elaborated the comparison with previous studies, and clarified the advantages of the full-coverage XCO2 product we generated:

"To validate the effectiveness of our model and resulting XCO2 products, we compared our results with current studies which focuses on global XCO2 reconstruction (Table 5). As for the in-situ validation, most existing studies report high accuracy with almost all R2 over 0.9, RMSE less than 2 ppm. Regarding spatial resolution, the various products differ substantially, ranging from 1° down to 0.01°. It should be noted that increasing spatial resolution tends to compromise the accuracy of XCO2 retrievals. However, our XCO2 product achieves an optimal balance between spatial detail and measurement precision, exhibiting both high spatial resolution (0.05°) and robust accuracy (R2=0.91, RMSE =1.54 ppm) in comprehensive evaluations.

**Table 5.** Comparison between current studies focusing on global XCO2 reconstruction

| Model                  | Spatial resolution | In-situ validation |       | ation | Reference           |
|------------------------|--------------------|--------------------|-------|-------|---------------------|
|                        |                    | (with TCCON)       |       |       |                     |
|                        |                    | $\mathbb{R}^2$     | RMSE  | MAE   | -                   |
|                        |                    |                    | (ppm) | (ppm) |                     |
| Attentional-based LSTM | 0.05°              | 0.91               | 1.54  | 1.22  | Our study           |
| Deep forest            | $0.1^{\circ}$      | 0.96               | 1.01  | -     | Zhang et al. (2023) |
| S-STDCT                | 0.25°              | 0.95               | 1.18  | -     | Wang et al. (2023)  |
| Spatiotemporal kriging | 1°                 | 0.97               | 1.13  | 0.88  | Sheng et al. (2022) |
| MLE & OI               | $0.5^{\circ}$      | 0.92               | 2.62  | 1.53  | Jin et al. (2022)   |
| ERT                    | 0.01°              | 0.83               | 1.79  | -     | Li et al. (2022)    |

\*S-STDCT: Self-supervised spatiotemporal discrete cosine transform, MLE & OI: maximum likelihood estimation method and optimal interpolation; ERT: Extremely randomized trees

To evaluate the advancement of our XCO2 product, we compared it with original OCO-2 observations and publicly available global XCO2 datasets (Wang et al., 2023;

Sheng et al., 2022; Zhang et al., 2023) across four regions: North America, Europe with northern Africa, Asia, and Oceania (Fig. 14) in January 2015. Despite monthly aggregation, OCO-2 data exhibit persistent spatial discontinuities, limiting the capacity to analyze monthly XCO2 variability at regional and national scales. Existing XCO2 products (spatial resolution of 0.25°, 1°, and 0.1°, respectively) broadly reproduce large-scale XCO2 patterns but fail to resolve fine-scale heterogeneity. In comparison, our reconstructed XCO2, with the highest spatial resolution, provides a more detailed and accurate representation of the regional XCO2 patterns. For example, lower XCO2 concentrations are clearly identified in eastern Canada (The first row of Fig. 14) and Papua New Guinea (The fourth row of Fig. 14), regions characterized by dense forest cover. This correspondence highlights the substantial carbon sink potential of these forested areas. Our high-resolution product better identifies the CO2 heterogeneity associated with different land cover types, whereas the coarse-resolution products smooth these signals. This limitation primarily stems from the neglect of highresolution and dependence on land cover dynamics coarse-resolution assimilated/reanalysis datasets (e.g., CAMS XCO2, CarbonTracker), resulting in oversmoothed spatial patterns that obscure satellite-derived high-resolution signals. Unlike assimilation-dependent approaches, our method avoids XCO2 reanalysis inputs, preserving satellite-scale fidelity through high-resolution environmental variables modeling while maintaining precision.

Figure 14. Comparison between the OCO-2 XCO2 data, accessible XCO2 products

from Wang et al. (2023), Sheng et al. (2022), Zhang et al. (2023), and our reconstructed XCO2 data in four regions, using the products of January of 2015 as an example." (Page 23-25 Line 504-542)

(2) Additionally, the global spatial distribution characteristics of XCO2 need more detailed discussion.

**Response:** Many thanks for this comment. We have added two figures and more description of the global spatial distribution characteristics of XCO2 in 3.2 **Spatiotemporal pattern of global XCO2** as follows:

"The global distribution of annual mean XCO2 concentration from 2015 to 2021 is illustrated in Fig. 9. The results reveal pronounced spatial heterogeneity in XCO2 concentrations, characterized by a marked hemispheric asymmetry. Specifically, the Northern Hemisphere exhibited systematically elevated XCO2 levels compared to the Southern Hemisphere, consistent with latitudinal gradients driven by anthropogenic emission patterns and atmospheric transport dynamics. Regionally, North America, East Asia, Central Africa, and northwest of Southern America were identified as persistent hotspots of enhanced XCO2. The high concentrations of XCO2 in North America and East Asia stem primarily from the fossil fuel emission from energy production and transportation sectors. Whereas the tropical regions (i.e., Central Africa and South America) are influenced by coupled biomass burning and land-use changes.

**Figure 9.** The global spatial distribution of reconstructed annual mean XCO2 concentration from 2015 to 2021.

We also provided the annual OCO-2 XCO2 data from 2015 to 2019 and OCO-3 XCO2 data from 2020 to 2021 in Fig. 10. Spatially, our reconstructed XCO2 dataset demonstrates robust consistency with satellite observations, particularly in mid-latitude industrialized regions where both datasets capture emission hotspots. Notably, OCO-3 exhibits denser observational sampling due to its improved spatial coverage and swath width compared to OCO-2's narrow tracks. However, persistent data gaps remain prevalent in both two satellite products after annual aggregating. These spatial coverage limitations hinder fine-scale global analysis, particularly in assessing localized emission sources and regional scale carbon flux."

**Figure 10.** The global spatial distribution of annual mean OCO-2/OCO-3 XCO2 concentration from 2015 to 2021.

**(Page 18-20 Line 404-429)**

**Q4.** (1) In the conclusion or discussion section, please clearly specify the concrete data or scientific significance of the high-resolution XCO2.

**Response:** Thanks for this constructive suggestion. We have provided further examples to introduce the significance of high-resolution XCO2 products, in **4.1 Comparison** with previous studies we supplemented the comparison with other coarse-resolution data products as follows:

"To evaluate the advancement of our XCO2 product, we compared it with original OCO-2 observations and publicly available global XCO2 datasets (Wang et al., 2023; Sheng et al., 2022; Zhang et al., 2023) across four regions: North America, Europe with northern Africa, Asia, and Oceania (Fig. 13) in January 2015. Despite monthly aggregation, OCO-2 data exhibit persistent spatial discontinuities, limiting the capacity to analyze monthly XCO2 variability at regional and national scales. Existing XCO2 products (spatial resolution of 0.25°, 1°, and 0.1°, respectively) broadly reproduce large-scale XCO2 patterns but fail to resolve fine-scale heterogeneity. In comparison,

our reconstructed XCO2, with the highest spatial resolution, provides a more detailed and accurate representation of the regional XCO2 patterns. For example, lower XCO2 concentrations are clearly identified in eastern Canada (The first row of Fig. 13) and Papua New Guinea (The fourth row of Fig. 13), regions characterized by dense forest cover. This correspondence highlights the substantial carbon sink potential of these forested areas. Our high-resolution product better identifies the CO2 heterogeneity associated with different land cover types, whereas the coarse-resolution products smooth these signals. This limitation primarily stems from the neglect of highresolution land cover dynamics and dependence on coarse-resolution assimilated/reanalysis datasets (e.g., CAMS XCO2, CarbonTracker), resulting in oversmoothed spatial patterns that obscure satellite-derived high-resolution signals. Unlike assimilation-dependent approaches, our method avoids XCO2 reanalysis inputs, preserving satellite-scale fidelity through high-resolution environmental variables modeling while maintaining precision."

**Figure 14.** Comparison between the OCO-2 XCO2 data, accessible XCO2 products from Wang et al. (2023), Sheng et al. (2022), Zhang et al. (2023), and our reconstructed XCO2 data in four regions, using the products of January of 2015 as an example.

**(Page 24-25 Line 517-542)**

And we also elaborated further in **6.** Conclusion as follows:

"As a major driver of global warming, the monitoring of CO2 changes, especially anthropogenic CO2 emissions, is of critical importance. The launch of carbon satellites

offers a significant advancement for CO2 monitoring. However, the limited spatial coverage of satellite observations constrains the utility of XCO2 data. While current XCO2 products exhibit relatively high validation accuracy, their coarse spatial resolution remains inadequate for applications such as regional- or county-level emission monitoring, as well as for the detection and inversion of large emission sources. To address these issues, we reconstructed a global full-coverage XCO2 product at a fine spatial resolution of 0.05° and temporal resolution of 1 month from 2015 to 2021. The advanced deep learning method was adopted to model time-series XCO2 and incorporate terrestrial flux, anthropogenic flux and climatic impacts into the parameterization process. Through comprehensive evaluations, including cross-validation, in-situ validation, spatial distribution assessment and comparison with other XCO2 products, our reconstructed XCO2 products demonstrates significant improvements in both accuracy and spatial resolution. The main conclusions and contributions are as following:

- (1) The advanced At-BiLSTM model could successfully established the nonlinear relationship between satellite-derived XCO2 and a set of key environmental variables. And the reconstructed XCO2 based on our model shows relatively good agreement with TCCON XCO2, with R2, RMSE, and MAE values of 0.91, 1.58 ppm, and 1.22 ppm, respectively.
- (2) The reconstructed XCO2 product overcomes the extensive data gaps typically caused by narrow satellite swaths and retrieval interference from clouds and aerosols, achieving complete global coverage. Moreover, relative to existing publicly available full-coverage XCO2 datasets, our product offers the finest spatial resolution (0.05°) while maintaining comparable accuracy.
- (3) Our method avoids coarse XCO2 reanalysis inputs, preserving satellite-scale fidelity through high-resolution environmental variables modeling. Consequently, the products enable enhanced ability in identifying regional- and county-level XCO2 hotpots, carbon emissions and fragmented carbon sinks, providing a robust basis for targeted global carbon governance policies." (Page 26-27 Line 580-610)
- (2) Additionally, provide an outlook for future research, outlining key issues to address in global XCO2 or CO2 concentration reconstruction studies, such as critical challenges or priorities that should be focused on.

**Response:** Thank you for this comment. We have added section **4.2 Limitations and future improvements** to provide a more detailed discussion of key challenges and the future outlook. Two key points are highlighted: firstly, the incorporation of auxiliary variables that capture vertical CO2 transport. Secondly, the enhancement of satellite observation coverage and accuracy to minimize data gaps and retrieval errors. The revised context is as follows:

"Additionally, though our model integrates multiple environmental variables associated with surface carbon flux variations, it does not account for vertical atmospheric transport. As XCO2 represents the column-averaged CO2 concentration, vertical redistribution of CO2 through atmospheric transport (e.g., mixing, convection) can alter the relationship between surface carbon fluxes and column concentrations. The absence of such vertical transport indicators may reduce the model's accuracy in regions or periods with strong vertical mixing. Future efforts will incorporate vertical transport-related variables, such as planetary boundary layer height, vertical wind components, and other reanalysis-derived indicators, to better represent the atmospheric processes that influence the column-averaged CO2 signal.

Moreover, while OCO missions currently provide some of the most accurate carbon satellite-based XCO2 retrievals, they still encounter some retrieval errors and data gaps driven by algorithmic limitations and variable meteorological conditions. A critical research frontier is the refinement of XCO2 retrieval algorithms to mitigate systematic biases in high-aerosol-load regions (e.g., industrial regions and biomass-burning plumes). Additionally, next-generation hyperspectral satellites, such as the upcoming CO2M (Copernicus Anthropogenic CO2 Monitoring Mission) with 2×2 km2 resolution and GeoCarb (Geostationary Carbon Observatory) offering hourly monitoring, will enhance spatial-temporal coverage and reduce cloud-induced data gaps (Reuter et al., 2025)." (Page 25-26 Line 553-572)

**Q5.** The construction of the OCO dataset is unclear. For instance, it is not specified how grids containing both OCO-2 and OCO-3 data within the same time period were processed.

**Response:** Thanks for this comment. In this study, given that OCO-3 has more intensive observations, we utilized the OCO-3 XCO2 data for all available year (i.e., 2020-2021) and used the data of OCO-2 for the other years (i.e., 2015-2019). Although

OCO-3 began providing data in August 2019, we used OCO-2 data for entire 2019 to maintain consistency in our monthly estimates.

In addition, analysing OCO-2 and OCO-3 data simultaneously may introduce several uncertainties due to their different spatiotemporal coverages. However, OCO-3 has a similar sensor with OCO-2 and inherits the retrieval algorithms of OCO-2. According to Taylor et al. (2023), the mean differences between OCO-3 and OCO-2 are around 0.2 ppm over land. Therefore, we suppose that the discrepancies between their datasets are minimal, and the combined analysis of data from these two satellites will have a negligible impact on our results. And we discussed this in **4.2 Limitations** and future improvements as follows:

"In terms of the satellite data, OCO-2 and OCO-3 provide different spatiotemporal coverages. Analyzing OCO-2 and OCO-3 data simultaneously may introduce several uncertainties due to these differences. However, OCO-3 has a similar sensor and inherits the retrieval algorithms of OCO-2. According to Taylor et al. (2023), the mean differences between OCO-3 and OCO-2 are around 0.2 ppm over land. Therefore, we suppose that the discrepancies between their datasets are minimal, and the combined analysis of data from these two satellites will have a negligible impact on our results." (Page 25 Line 545-552)

**Q6.** The study utilized various satellite-derived variables, including land flux, anthropogenic flux, and climatic impacts, for global XCO2. However, it is unclear whether these satellite data have gaps, particularly in high-latitude regions. If gaps exist, the study should specify how they were addressed.

Response: Many thanks for this constructive comment. Among all ancillary variables, those related to climatic impacts from ERA5-Land, as well as land use and cover change (LUCC), vegetation continuous fields (VCF), and nighttime lights (NTL), provide full spatial coverage. Given that the XCO2 reconstruction was performed on a monthly scale, all satellite-derived variables were aggregated to monthly averages using Google Earth Engine (GEE). During this averaging process, most data gaps were effectively filled. For variables that still contained missing values after monthly aggregation, we applied bilinear interpolation methods to fill the remaining gaps.

We have added the description of data processing in **2.1.3 Environmental** variables as follows:

"All data were converted to monthly time-series. The bilinear interpolation approach was employed both to fill gaps in the ancillary data and to convert the data at different spatial resolutions to 0.05° resolution." (Page 10 Line 252-254)

**Q7.** Line 243-244. "...spatial resolutions to 1 km resolution". The '1 km resolution' is inconsistent with the study's focus on a 0.05°

**Response:** Thank you for your correction. We apologize for the typographical error and all data have been processed at the resolution of 0.05° to match the XCO2 products. We have revised these accordingly.

**Q8.** The manuscript contains several typo errors. For instance, in Line 303, "Figure 5. (a) Density scatterplots of sample-based..." includes an unnecessary '(a)'. In Line 314, the value '1.21' should be corrected to '1.22 ppm'. These errors should be revised for accuracy and clarity.

**Response:** Thank you for the correction. We have revised these accordingly and checked the full manuscript to avoid such typo errors.